# Compensatory growth renders Tcf7l1a dispensable for eye formation despite its requirement in eye field specification

Rodrigo M Young[1‡*], Thomas A Hawkins[1†], Florencia Cavodeassi[1†§], Heather L Stickney[1#], Quenten Schwarz[1¶], Lisa M Lawrence[1**], Claudia Wierzbicki[1††], Bowie YL Cheng[1‡‡], Jingyuan Luo[1], Elizabeth Mayela Ambrosio[1], Allison Klosner[1], Ian M Sealy[2,3], Jasmine Rowell[1], Chintan A Trivedi[1], Isaac H Bianco[1], Miguel L Allende[4], Elisabeth M Busch-Nentwich[2,3], Gaia Gestri[1], Stephen W Wilson[1*]

[1]Department of Cell and Developmental Biology, University College London, London, United Kingdom; [2]Wellcome Sanger Institute, Wellcome Genome Campus, Hinxton, United Kingdom; [3]Department of Medicine, University of Cambridge, Cambridge, United Kingdom; [4]Center for Genome Regulation, Facultad de Ciencias, Universidad de Chile, Santiago, Chile

*For correspondence:
rodrigo.young@ucl.ac.uk (RMY);
s.wilson@ucl.ac.uk (SWW)

†These authors contributed equally to this work

Present address: ‡Institute of Ophthalmology, University College London, London, United Kingdom; §Institute of Medical and Biomedical Education, St George's University of London, London, United Kingdom; #Ambry Genetics, Aliso Viejo, United States; ¶Centre for Cancer Biology, University of South Australia and SA Pathology, Adelaide, Australia; **Department of Molecular Medicine, University of Auckland School of Medicine, Auckland, New Zealand; ††Department of Infectious Disease Epidemiology, MRC Centre for Global Infectious Disease Analysis, School of Public Health, Imperial College London, London, United Kingdom; ‡‡Department of Medicine, The University of Hong Kong, Hong Kong, China

Competing interests: The authors declare that no competing interests exist.

**Abstract** The vertebrate eye originates from the eye field, a domain of cells specified by a small number of transcription factors. In this study, we show that Tcf7l1a is one such transcription factor that acts cell-autonomously to specify the eye field in zebrafish. Despite the much-reduced eye field in *tcf7l1a* mutants, these fish develop normal eyes revealing a striking ability of the eye to recover from a severe early phenotype. This robustness is not mediated through genetic compensation at neural plate stage; instead, the smaller optic vesicle of *tcf7l1a* mutants shows delayed neurogenesis and continues to grow until it achieves approximately normal size. Although the developing eye is robust to the lack of Tcf7l1a function, it is sensitised to the effects of additional mutations. In support of this, a forward genetic screen identified mutations in *hesx1*, *cct5* and *gdf6a*, which give synthetically enhanced eye specification or growth phenotypes when in combination with the *tcf7l1a* mutation.
DOI: https://doi.org/10.7554/eLife.40093.001

## Introduction

The paired optic vesicles originate from the eye field, a single, coherent group of cells located in the anterior neural plate (*Cavodeassi, 2018*). During early neural development, the specification and relative sizes of prospective forebrain territories, including the eye field, depend on the activity of the Wnt/β-Catenin and other signalling pathways (*Beccari et al., 2013*; *Cavodeassi, 2014*; *Wilson and Houart, 2004*). Enhanced Wnt/β-Catenin activity leads to embryos with small or no eyes (*Cavodeassi et al., 2005*; *Kim et al., 2000*; *Heisenberg et al., 2001*; *Houart et al., 2002*). In contrast, decreasing activity of Wnt/β-Catenin signalling generates embryos with bigger forebrain and eyes (*Cavodeassi et al., 2005*; *Glinka et al., 1998*; *Lekven et al., 2001*; *Houart et al., 2002*). Although much research has focused on the molecular mechanisms involved in the specification of the eye field, little is known about what happens to the eyes if eye field size is disrupted.

A number of genes have been identified as encoding a transcription factor network that specifies the eye field (*Beccari et al., 2013*; *Zuber et al., 2003*). These genes have been defined based on conserved cross species expression patterns in the anterior neuroectoderm and on phenotypes observed when overexpressed or when function is compromised (*Beccari et al., 2013*). Perhaps

**eLife digest** Left and right eyes develop independently, yet they consistently grow to roughly the same size in humans and other creatures. How they do this remains a mystery, though scientists have learned that both eyes originate from a single group of cells in the developing nervous system called the eye field. As development progresses, the eye field splits in two, and buds into the two separate compartments from which each eye forms. As the eyes grow, the cells in each compartment specialize, or 'differentiate', to make working left and right eyes. Scientists often study eye development in zebrafish embryos because it is easy to see each step in the process.

Now, Young at al. show that zebrafish with a mutation that causes the eye field to be half its normal size go on to form normal-sized eyes. Somehow these developing embryos overcome this deleterious mutation. It turns out that the eyes of zebrafish with this mutation grow for a longer period of time than typical zebrafish eyes. This change allows the mutant fish's eyes to catch up and reach normal size. When Young et al. removed some cells from one of the forming eyes of normal zebrafish embryos they found that same thing happened. The smaller eye developed for a longer time and delayed its differentiation until both eyes were the same size. Conversely, when eyes developed from a larger than normal eye field, growth stopped prematurely and differentiation began early preventing the eyes from ending up oversized.

Though the fish were able to overcome the effects of one mutation to develop normal-sized eyes, adding a second mutation that affected eye development led to unusual sized eyes or absence of eyes. Together the experiments identify genes and mechanisms essential for the formation and size of the eyes. Given that the processes underlying eye formation are very similar in many animals, this new information should help scientists to better understand eye abnormalities in humans.
DOI: https://doi.org/10.7554/eLife.40093.002

surprisingly, to date, there are relatively few mutations that lead to complete loss of eyes suggesting that early stages of eye development are robust to compromised function of genes involved in eye development. Indeed, in humans, eye phenotypes are often highly variable in terms of penetrance and expressivity even between left and right eyes (*Reis and Semina, 2015*; *Williamson and FitzPatrick, 2014*). This again raises the possibility that the developing eye is robust and can sometimes cope with mutations in genes involved in eye formation.

Genetic robustness is the capacity of organisms to withstand mutations, such that they show little or no phenotype, or compromised viability (*Félix and Barkoulas, 2015*; *Wagner, 2005*). This inherent property of biological systems is wired in the genetic and proteomic interactomes and enhances the chance of viability of individuals in the face of mutations. High-throughput reverse mutagenesis projects and the emergence of CRISPR/Cas9 gene editing techniques have highlighted the fact that homozygous loss of function mutations in many genes generate viable mutants with no overt phenotype (*Varshney et al., 2015*; *Dickinson et al., 2016*; *Meehan et al., 2017*). Across phyla, mutations in single genes are more likely to give rise to viable organisms than to show overt or lethal phenotypes. For instance, it is estimated that zygotic homozygous null mutations in just ~7% of zebrafish genes compromise viability before 5 days post-fertilisation (*Kettleborough et al., 2013*) and 8–10% between day 5 and 3 months (Shawn Burgess, personal communication); and compromised viability is predicted following loss of function for about 35% of mice genes (*Dickinson et al., 2016*; *Meehan et al., 2017*). Furthermore, apparently healthy viable homozygous or compound heterozygous 'gene knockouts' have been found for 1171 genes in the Icelandic human population (*Sulem et al., 2015*) and for 1317 genes in the Pakistani population (*Saleheen et al., 2017*).

In some cases, the lack of overt phenotype may be due to redundancy in gene function based on functional compensation by paralogous or related genes (*Barshir et al., 2018*; *Hurles, 2004*; *Wagner, 1996*). We can assume that genes that do not express a phenotype when mutated are not lost to genetic drift because in some way they enhance the fitness of the species. For instance, even though two paralogous *Lefty* genes encoding Nodal signalling feedback effectors have been shown to be individually dispensable for survival, they do make embryonic development robust to signalling noise and perturbation (*Rogers et al., 2017*).

Genetic compensation for deleterious mutations is a cross-species feature (*El-Brolosy and Stainier, 2017*), and mRNAs that undergo nonsense-mediated decay due to mutations that lead to premature termination codons can upregulate the expression of paralogous and other related genes (*El-Brolosy et al., 2018*). However, only a fraction of genes have paralogues and other compensatory mechanisms must contribute to the ability of the embryo to cope with potentially deleterious mutations. One such mechanism is distributed robustness, which can emerge in gene regulatory networks (*Wagner, 2005*). This kind of robustness relies on the ability of the network to regulate the expression of genes and/or the activity of proteins within the network, such that homeostasis is preserved when one of its components is compromised (*Davidson, 2010*; *Peter and Davidson, 2016*).

Maternal-zygotic *tcf7l1a* mutant zebrafish have been previously described as lacking eyes (*Kim et al., 2000*). In this study, we show that expression of this phenotype is dependent on the genetic background. We find that *tcf7l1a* mutants can develop functional eyes and are viable, and that this is not due to compensatory upregulation of other *lef/tcf* genes. Despite the presence of functional eyes, the eye field in *tcf7l1a* mutants is only half the size of the eye field of wildtype embryos, indicating an early requirement for *tcf7l1a* during eye field specification. We further show that this requirement is cell autonomous, revealing a striking dissociation between the early role and requirement for Tcf7l1a in eye field specification and the later absence of an overt eye phenotype. Subsequent to compromised eye field specification, *tcf7l1a* mutant eyes recover their size by delaying neurogenesis and prolonging growth in comparison to wildtype eyes. This compensatory ability of the developing eye was also observed when cells were removed from wild-type optic vesicles. Altogether, our study suggests that the loss of Tcf7l1a does not trigger any genetic compensation or signalling pathway changes that restore eye field specification; instead, the developing optic vesicle shows a remarkable ability to subsequently modulate its development to compensate for the early, severe loss of eye field progenitors.

The penetrance and expressivity of eye phenotypes appears to be dependent on complex genetic and environmental interactions (*Gestri et al., 2009*; *Kaukonen et al., 2018*; *Prokudin et al., 2014*). Thus, we speculated that *tcf7l1a* mutant eyes may be sensitised to the effects of additional mutations. Here, we show this is indeed the case and describe the isolation of three mutations from a recessive synthetic modifier screen in *tcf7l1a* homozygous mutant zebrafish that lead to enhanced/novel eye phenotypes when in combination with loss of *tcf7l1a* function.

In summary, our work shows that zebrafish eye development is robust to the effects of a mutation in *tcf7l1a* due to compensatory growth mechanisms that may link eye size and neurogenesis. Our study adds to a growing body of research revealing a variety of mechanisms by which the developing embryo can cope with the effects of deleterious genetic mutations.

## Results

### The *tcf7l1a*^m881/m881^ mutation is fully penetrant but maternal-zygotic mutants show no overt eye phenotype and are viable

The *headless (hdl)*^m881^ mutation in *tcf7l1a* (*tcf7l1a*^-/-^ from here onwards) was identified because embryos lacking maternal and zygotic (MZ) gene function lacked eyes (*Kim et al., 2000*). However, no overt defects were observed in zygotic (Z) *tcf7l1a*^-/-^ mutants, due to functional redundancy with the paralogous *tcf7l1b* gene (*Dorsky et al., 2003*). In our facility, *MZtcf7l1a*^-/-^ embryos initially showed a variable eye phenotype, ranging from eyeless, to small and overtly normal eyes, with proportions that varied in clutches from different pairs of fish (not shown). We hypothesised that genetic background effects could be responsible for either enhancing or suppressing the eyeless phenotype. To test this idea, we outcrossed *tcf7l1a*^-/-^ fish to *ekkwill* (EKW) or AB wildtype fish and identified *tcf7l1*^+/-^ carriers by PCR genotyping. After three generations of outcrossing to EKW or AB fish, we incrossed *tcf7l1*^+/-^ carriers to grow Z*tcf7l1a*^-/-^ adults. All *MZtcf7l1a*^-/-^ embryos coming from six pairings of Z*tcf7l1a*^-/-^ mutant fish developed eyes only slightly reduced in size compared to eyes of wildtype embryos of the same EKW or AB strain (100%, n > 100; *Figure 1A,B*).

The *tcf7l1a*^m881^ mutation creates a splice acceptor site in intron 7, which leads to a seven nucleotide insertion in *tcf7l1a* mRNA that gives rise to a truncated protein due to a premature termination codon (*Kim et al., 2000*). Given that the wildtype splice site in intron seven is still present in *tcf7l1a* mutants, we assessed whether the lack of phenotype in *MZtcf7l1a*^-/-^ mutants could be due to

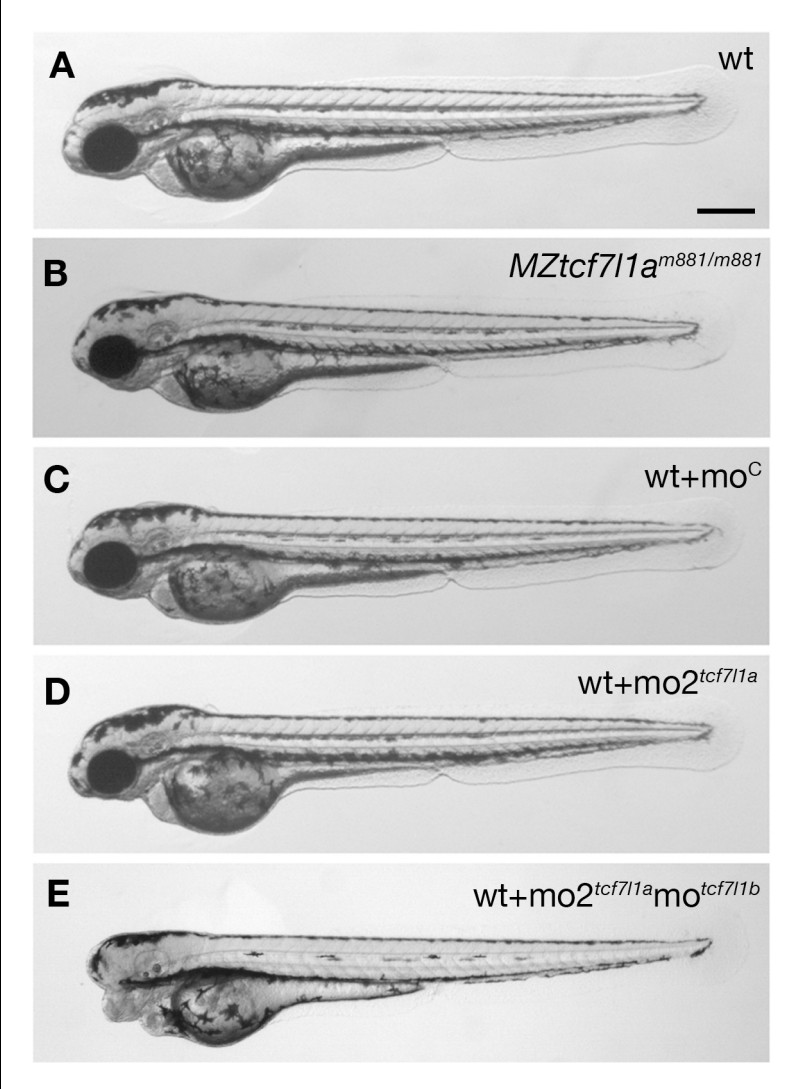

**Figure 1.** *Tcf7l1a* maternal zygotic (MZ) mutants and *tcf7l1a* morphants have no overt eye phenotype. Lateral views of typical wildtype (**A**) *MZtcf7l1a*$^{-/-}$ (**B**) wildtype injected with control morpholino (**C**) *tcf7l1a* morphant (**D**) and *tcf7l1a/tcf7l1b* double morphant (**E**) embryos at 2 days post fertilisation. All conditions n > 100 and over three independent experiments except when specified. Dorsal up, anterior to the left. Scale Bar = 250 μm.
DOI: https://doi.org/10.7554/eLife.40093.003

The following figure supplements are available for figure 1:

**Figure supplement 1.** Sequence of the *tcf7l1a* exon7/8 boundary.
DOI: https://doi.org/10.7554/eLife.40093.004

**Figure supplement 2.** Alignment of *tcf7l1a*, morpholinos to *lef1* and other *tcf* genes.
DOI: https://doi.org/10.7554/eLife.40093.005

incomplete molecular penetrance as a result of expression of mRNA from both wildtype and mutant splice sites. The chromatogram sequence of the RT-PCR product amplifying exons 7 and 8 in wild-type, mutant and heterozygous embryos showed that only wildtype *tcf7l1a* mRNA was detected in wildtype embryos and only mutant mRNA containing the seven nucleotide insertion was observed in mutants, while heterozygous embryos produced both wildtype and mutant mRNAs (*Figure 1—figure supplement 1*; *Kim et al., 2000*). This suggests that the mutant splice site is the only one used in *tcf7l1a*$^{-/-}$ embryos. In addition, while overexpression of wildtype *tcf7l1a* mRNA rescued eye formation in embryos in which both *tcf7l1a* and *tcf7l1b* are knocked down, *tcf7l1a*$^{m881}$ mutant mRNA did not, confirming that protein arising from the *tcf7l1a*$^{m881}$ allele is not functional (not shown;

*Kim et al., 2000*). These observations suggest that the *m881* allele is indeed a null mutation and that *tcf7l1a* is not essential for eye formation.

Supporting a requirement for *tcf7l1a* to form eyes, antisense morpholino knockdown of *tcf7l1a* (mo1*tcf7l1a*) leads to eyeless embryos (*Dorsky et al., 2003*) comparable to the originally described *headless MZtcf7l1a-/-* mutant phenotype (*Kim et al., 2000*). However, the target site for the morpholino used in that study shows considerable sequence homology to the translation start ATG region of other *tcf* gene family members (56–76%; *Figure 1—figure supplement 2A*). This suggests that the mo1*tcf7l1a* phenotype may be due to the morpholino knocking down expression of other *tcf* genes, as has been described for other morpholinos targeting paralogous genes (*Kamachi et al., 2008*). Indeed, injection of a different *tcf7l1a* morpholino (mo2*tcf7l1a*) with low homology to other *tcf* genes (36–45%, *Figure 1—figure supplement 2B*) did not lead to an eyeless phenotype (0.4 pMol/embryo, 100%, n > 100; *Figure 1C,D*). *tcf7l1b* morpholino injection on its own showed no overt phenotype (*Dorsky et al., 2003*) but co-injection of mo2*tcf7l1a* and mo*tcf7l1b* gave rise to eyeless embryos (each at 0.2 pMol/embryo, 78.26%, n = 92, over three experiments; *Figure 1E* and see *Dorsky et al., 2003*). This suggests that even though mo2*tcf7l1a* injection alone resulted in no phenotype, the morpholino does knockdown *tcf7l1a*.

Together, these results suggest that even though *tcf7l1a-/-* is a fully penetrant null mutation, lack of maternal and zygotic *tcf7l1a* function alone does not lead to loss of eyes in all genetic backgrounds.

## *tcf7l1a* loss of function is not compensated by upregulation of other *tcf* genes

Z*tcf7l1a-/-* and MZ*tcf7l1a-/-* embryos develop eyes, whereas embryos lacking both Z*tcf7l1a* and Z*tcf7l1b* do not (*Dorsky et al., 2003*). Thus, we hypothesised that enhanced expression of the paralogous *tcf7l1b*, or other *lef/tcf* genes may compensate for the absence of *tcf7l1a* function, as shown for other mutations (*El-Brolosy et al., 2018*; *Rossi et al., 2015*). To test this idea, we assessed the expression of all *lef/tcf* genes by RT-qPCR in sibling wildtype and Z*tcf7l1a-/-* mutant embryos at the stage when the eye field has been specified (10 hr post-fertilisation; hpf).

Expression levels of *lef/tcf* genes did not increase in Z*tcf7l1a-/-* mutant embryos suggesting that there is no compensatory upregulation (*Figure 2A*, *Supplementary file 1A*). As previously shown, *tcf7l1a* undergoes nonsense-mediated decay in mutants resulting in reduced expression levels (*Kim et al., 2000*; *Figure 2A*; *Supplementary file 1A*). *lef1* and *tcf7* levels did not change significantly in mutants and *tcf7l1b* (*tcf3b*) and *tcf7l2* (*tcf4*) expression was actually reduced to 63 ± 6% and 62 ± 8% respectively of wildtype levels (*Figure 2A*; *Supplementary file 1A*). The *otx1b* and *otx2* genes, which are expressed in the anterior neural plate, also showed slightly reduced expression (*otx1b*, reduced to 81 ± 11% and *otx2*, 79 ± 10%) suggesting that the anterior neural plate may be slightly reduced in size in *tcf7l1a* mutants. Indeed, the domain of the neural plate encompassed by expression of *emx3* around the anterior margin of the neural plate up to the mesencephalic marker *pax2a* (*Figure 2D,E*) was reduced to 76% of wildtype size in *tcf7l1a* mutants (n = 11, p=0.0041, *Figure 2B*; *Supplementary file 1B*). This indicates that a reduction in the size of the prospective forebrain of Z*tcf7l1a-/-* embryos may contribute to the reduced levels of expression of *tcf7l1b*, *tcf7l2* and *otx* genes.

Overall, these results suggest that *tcf* genes do not show compensatory regulation in response to loss of *tcf7l1a* function.

## Optic vesicles evaginate and form eyes in MZ*tcf7l1a-/-* mutants despite a much-reduced eye field

More remarkable than the modest changes in *tcf* and *otx* gene expression was the finding that RT-qPCR showed very reduced expression of eye field genes in Z*tcf7l1a-/-* mutant embryos (*Figure 2A*; *rx3* reduced to 26 ± 1%, p=0.0002 and *six3b* reduced to 44 ± 5%, p=0.0091 of wildtype levels). Consequently, the presence of overtly normal looking eyes in both Z*tcf7l1a-/-* and MZ*tcf7l1a-/-* embryos is surprising given that *rx3-/-* mutant embryos lack eyes due to impaired specification/evagination of the optic vesicles (*Loosli et al., 2003*; *Stigloher et al., 2006*). We confirmed that expression of *six3b* and *rx3* is reduced in the anterior neural plate by *in situ* hybridisation in Z*tcf7l1a-/-* and *tcf7l1a* morphant embryos (100%,n > 40; *Figure 2F–I*; *Figure 2—figure supplement 1A,B*; similar changes

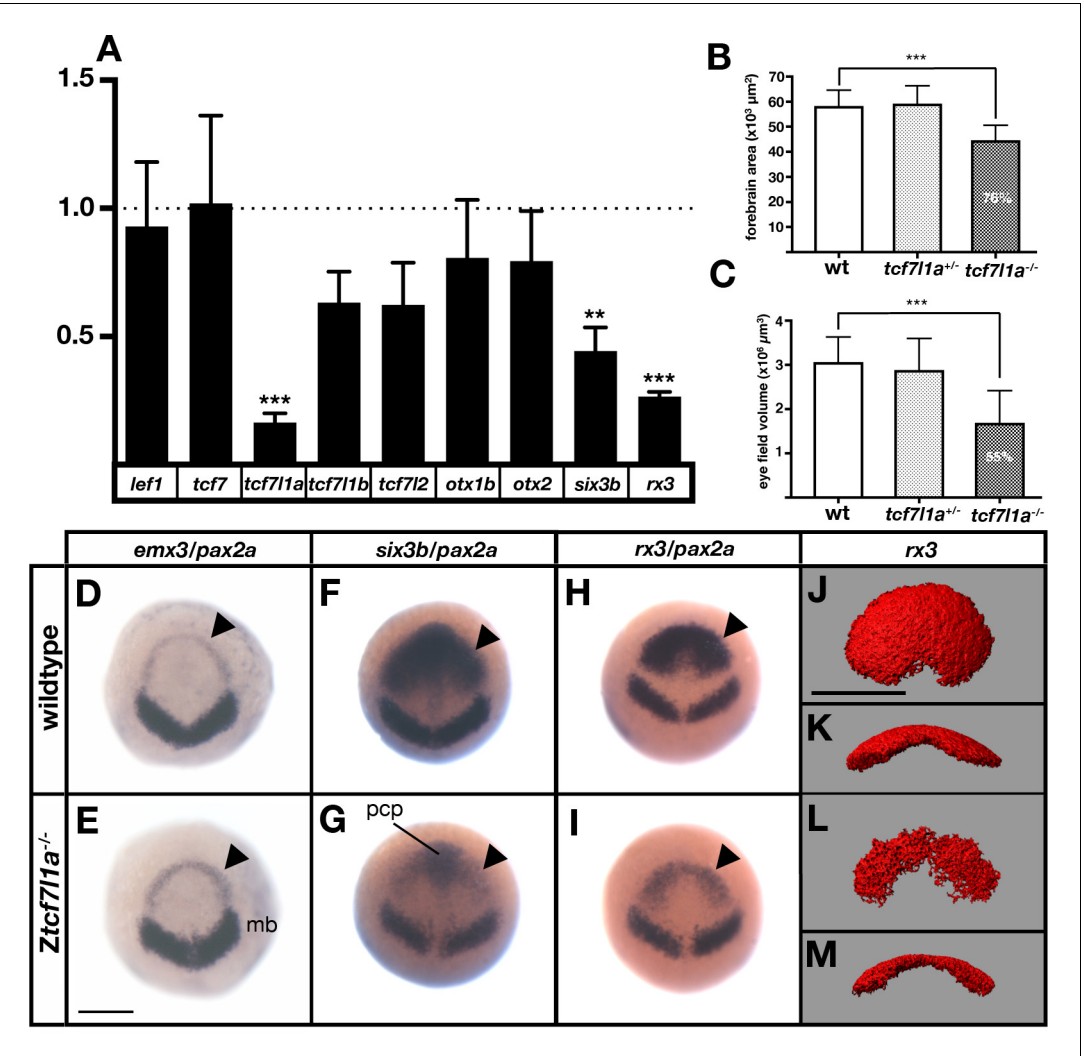

**Figure 2.** The prospective forebrain and eye field domains of the neural plate are reduced in Z*tcf7l1a*$^{-/-}$ mutants. (A) Graph showing RT-qPCR quantification of the mRNA levels of *lef1*, *tcf7*, *tcf7l1a*, *tcf7l1b*, *tcf7l2*, *otx1b*, *otx2*, *six3b* and *rx3* in Z*tcf7l1a*$^{-/-}$ mutants relative to wildtype embryos at 10hpf. Biological and technical triplicates, two independent experiments. (B, C) Quantification of the forebrain domain of the anterior neural plate (B) enclosed by *emx3* up to *pax2a* (D, E) expression by *in situ* hybridisation (reduction to an average of 76%, n = 11, one experiment, data in *Supplementary file 1B*), and eye field volume (C) by *rx3* fluorescent *in situ* hybridisation confocal volume reconstruction (J–M) (reduction to an average of 55%, n = 10, one experiment, data in *Supplementary file 1C*). (D–I) Expression of *emx3* (arrowhead)/*pax2a* (D, E), *six3b* (arrowhead)/*pax2a* (F, G) and *rx3* (arrowhead)/*pax2a* (H, I) in wildtype (D, F, H) and Z*tcf7l1a*$^{-/-}$ (E, G, I) embryos detected by *in situ* hybridisation at 10hpf. Reduction of *six3b* and *rx3* expression 100%, n > 40, three experiments. (J–M) Confocal volume reconstruction of *rx3* fluorescent *in situ* hybridisation in wildtype (J, K) and Z*tcf7l1a*$^{-/-}$ (L, M) mutants at 10hpf. (J, L) Dorsal view, anterior to top, and (K, M) transverse view from posterior, dorsal up. Abbreviations: mb, midbrain; pcp, prechordal plate Scale Bars = 250 µm.

DOI: https://doi.org/10.7554/eLife.40093.006

The following figure supplements are available for figure 2:

**Figure supplement 1.** *tcf7l1a* morpholino (mo2$^{tcf7l1a}$) phenocopies the Z*tcf7l1a*$^{-/-}$ mutant.

DOI: https://doi.org/10.7554/eLife.40093.007

**Figure supplement 2.** The eye field domain of the anterior neural plate is caudalised in Z*tcf7l1a* mutants.

DOI: https://doi.org/10.7554/eLife.40093.008

**Figure supplement 3.** Eye vesicle evagination in heterozygous and Z*tcf7l1a*$^{-/-}$ mutants.

DOI: https://doi.org/10.7554/eLife.40093.009

seen in MZ*tcf7l1a*$^{-/-}$ mutants, data not shown). The expression of *six3b* was reduced in the eye field but not in the prechordal plate of Z*tcf7l1a*$^{-/-}$ mutants, likely explaining why RT-qPCR showed a greater reduction in *rx3* than *six3b* expression (**Figure 2F–I**; **Supplementary file 1A**). Analysis of eye field volume by fluorescent *in situ* hybridisation of *rx3* revealed a reduction to 54.7% of wildtype size (n = 10, **Figure 2C,J–M**; **Supplementary file 1C**) and intensity of expression within the reduced eye field also appeared reduced (**Figure 2H,I**).

Further *in situ* hybridasation analysis suggests that it is the caudal region of the eye field that is most affected in Z*tcf7l1a*$^{-/-}$ mutants. *emx3* expression directly rostral to the eye field was slightly broader in Z*tcf7l1a*$^{-/-}$ mutants than wildtypes but expression did not encroach into the reduced eye field (**Figure 2D,E**; **Figure 2—figure supplement 2A,B**, n = 5 each condition). Conversely, expression of the prospective diencephalic marker *barhl2* caudal to the reduced eye field was expanded rostrally at 10hpf (**Figure 2—figure supplement 2C,D** n = 5 each condition) and even more evidently at 9hpf (**Figure 2—figure supplement 2E,F**, 13/13 Z*tcf7l1a*$^{-/-}$). These observations suggest a caudalisation of the anterior neural plate in Z*tcf7l1a*$^{-/-}$ mutants leading to reduced eye field specification consistent with phenotypes observed in conditions in which Wnt pathway repression is reduced (**Heisenberg et al., 2001**; **van de Water et al., 2001**).

RNAseq analysis of wildtype, Z*tcf7l1a*$^{-/-}$ and Z*tcf7l1a*$^{-/-}$/Z*tcf7l1b*$^{+/-}$ embryos at 8.5hpf (80–90% epiboly stage), when the eye field is first specified confirmed and extended RT-qPCR and *in situ* hybridisation analyses (**Supplementary file 1D**). In Z*tcf7l1a*$^{-/-}$, *hesx1* (**Kazanskaya et al., 1997**), *rx3*, *tcf7l1a*, and *fezf2* (**Sun et al., 2006**) which are expressed in the prospective forebrain/eyefield were downregulated, whereas *her5* and *irx1b* which are expressed more caudally in the neural plate (**Müller et al., 1996**, **Wang et al., 2001b**) were upregulated consistent with mild caudalisation of the neural plate. Additionally, in Z*tcf7l1a*$^{-/-}$/*tcf7l1b*$^{+/-}$ embryos, which to not form eyes, the expression of *tcf7l1b* was enhanced by about 40%, suggesting transcriptional compensation but evidently this was not sufficient to rescue eye formation.

Despite the small size of eye field in *tcf7l1a*$^{-/-}$ mutants, optic vesicles appear to evaginate normally. Time lapse analysis of optic vesicle evagination using the *Tg(rx3:GFP)*$^{zf460Tg}$ transgene to visualise eye field cells (**Brown et al., 2010**) showed that optic vesicle morphogenesis in Z*tcf7l1a*$^{-/-}$ embryos proceeded as in heterozygous sibling embryos (**Figure 2—figure supplement 3A,B**; *tcf7l1a*$^{+/-}$, n = 6 and Z*tcf7l1a*$^{-/-}$ n = 6; **Video 1** and **Video 2**).

## Tcf7l1a functions cell-autonomously to promote eye field specification

Although Tcfs regulate the balance between activation and repression of the Wnt/βCatenin pathway during anterior neural plate regionalisation (**Kim et al., 2000**; **Dorsky et al., 2003**), it is unclear if Tcf function is required for cells to adopt the eye field fate. To address this, we determined whether Tcf7l1a function is required cell-autonomously during eye formation by transplanting wildtype and MZ*tcf7l1a*$^{-/-}$ GFP labelled (GFP+) cells into wildtype and *tcf7l1a* mutant hosts and analysing the expression of *rx3* when eye specification has occurred (10hpf, 100% epiboly; **Figure 3**).

Transplants of wildtype cells to MZ*tcf7l1a*$^{-/-}$ mutant embryos led to the recovery of *rx3* expression exclusively restricted to the wildtype GFP+ cell clones (13/13 transplants, **Figure 3A–C**). However, the border of the GFP+ wildtype clones showed less *rx3* expression, suggesting that cells at the edge of the graft are subject to cell non-autonomous signalling effects from cells surrounding the clone. Conversely, MZ*tcf7l1a*$^{-/-}$ mutant GFP+ cells expressed much lower levels of *rx3* than wildtype neighbours when positioned in the eye field of wildtype embryos (9/9 transplants, **Figure 3D–F**). The reduction in *rx3* expression was limited to the MZ*tcf7l1a*$^{-/-}$ GFP+-mutant cells. Control transplants of cells from wildtype donor embryos to wildtype hosts showed no effect on *rx3* expression (not shown). Consistent with a cell autonomous role for Tcf7l1a in eye formation, overexpression of the

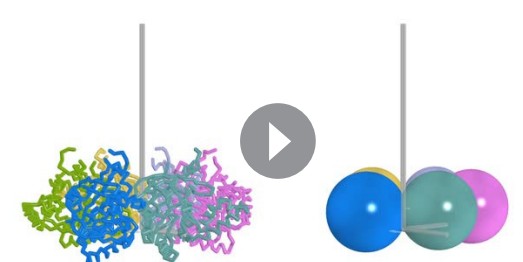

**Video 1.** Time lapse movies of eye vesicle evagination in and Z*tcf7l1a*$^{+/-}$ siblings. Confocal time lapse movies (1 frame every 5 min) of Z*tcf7l1a*$^{+/-}$ sibling (S1, n = 5) expressing the *Tg(rx3:GFP)*$^{zf460Tg}$ transgene. First frame taken at 11hpf; membrane RFP in red counterstain.
DOI: https://doi.org/10.7554/eLife.40093.010

Wnt inhibitor Dkk1 (*Hashimoto et al., 2000*) expanded the anterior neural plate in both wildtype and *tcf7l1a* mutants, but *rx3* expression and eye field size remained much smaller in the enlarged anterior plate of *tcf7l1a* mutants (*Figure 3G–J*).

All together, these results support a cell-autonomous role for Tcf7l1a in promoting eye field specification.

## Eye size in *Ztcf7l1a⁻/⁻* embryos recovers with growth kinetics similar to wildtype embryos

Despite a much-reduced eye field, eyes in Z*tcf7l1a⁻/⁻* fry and adults seem indistinguishable from those in wildtype siblings. Indeed, optokinetic responses of Z*tcf7l1a⁻/⁻* and wildtype 5dpf larvae showed no significant differences at any of the four tested spatial frequencies (*Figure 4—figure supplement 1*, *Supplementary file 1E*), suggesting that by this stage, Z*tcf7l1a⁻/⁻* eyes are functional and have visual acuity comparable to that of wildtype siblings. Consequently, although Z*tcf7l1a⁻/⁻* embryos show a robust and severe neural plate patterning phenotype, eye formation recovers over time. To explore how this recovery happens, we measured eye size in Z*tcf7l1a⁻/⁻* embryos from 24 to 96hpf (*Figure 4A,C–L*; *Supplementary file 1F*), estimating eye volumes from retinal profiles (see Materials and methods).

At 24hpf, eye volume in mutants was about 57% of the estimated volume of wildtype eyes at the same stage (*Figure 4A,C,H*; *Supplementary file 1F*). However, by 48hpf mutant eyes were about 85% of the size of eyes in wildtype/heterozygous siblings (*Figure 4A,G,L*; *Supplementary file 1F*). Z*tcf7l1a⁻/⁻* eye size did not further recover beyond this time point and up to 5dpf (*Figure 4B*). Eye growth in both wildtypes and Z*tcf7l1a⁻/⁻* mutants showed similar growth kinetics (*Figure 4A*). This suggests that even though Z*tcf7l1a⁻/⁻* eyes are smaller, they follow a comparable developmental time-course as wildtype eyes in the early growth phase between 24 and 36hpf but with about 8 hr delay (for instance, a 32hpf Z*tcf7l1a⁻/⁻* eye is about the same size as a wild-type 24hpf eye).

The temporal shift in eye growth in Z*tcf7l1a⁻/⁻* mutants is not explained by an overall developmental delay as the position of the posterior lateral line primordium (pLLP) was similar to wildtype at all stages tested (*Figure 4—figure supplement 2*, *Supplementary file 1G*). Volumes of eye cells in Z*tcf7l1a⁻/⁻* mutants and siblings were not significantly different at 24 or 36hpf and consequently cell size changes likely play no role in eye size compensation in Z*tcf7l1a* mutants (*Figure 4—figure supplement 3*, *Supplementary file 1H*).

## Eye size recovers after physical removal of optic vesicle cells

To assess if size recovery is a general feature of eye development, we physically removed optic vesicle cells in wildtype embryos and assessed the effect on eye growth. Cells were aspirated from one of the two nascent optic vesicles at 12hpf (six somite stage), leaving approximately the medial half of the vesicle intact (*Figure 4M*). At 36hpf, there was still a clear size difference between the experimental and control eyes (*Figure 4N*, *Figure 4—figure supplement 4*, *Supplementary file 1I*). However, by 4dpf we observed little or no size difference between control and experimental eyes (three independent experiments, n=20/20, *Figure 4O*, *Figure 4—figure supplement 4*, *Supplementary file 1I*). Three eyes from partially ablated optic vesicles which were ~90% the size of their control contralateral eyes at 30hpf, recovered to 100% by 3dpf (*Figure 4—figure supplement 4K*, *Supplementary file 1I*) and five eyes which were between ~65 and~75% of control size at 30hpf recovered to 90% by 3dpf with little or no further recovery by 4dpf (*Figure 4—figure supplement 4A–I,K*, *Supplementary file 1I*). Consequently, the forming eye can effectively recover from either genetic or physical reduction in the size of the eye field/evaginating optic vesicle.

## Neurogenesis is delayed in *tcf7l1a* mutant eyes

The observation that wildtype and Z*tcf7l1a⁻/⁻* mutant eyes display similar, but temporally offset, growth kinetics led us to speculate that that retinal neurogenesis might be delayed in Z*tcf7l1a⁻/⁻* eyes to extend the period of proliferative growth prior to retinal precursors undergoing neurogenic divisions.

In the zebrafish eye, neurogenesis can be visualised by tracking expression of *atoh7* (*ath5*) in retinal neurons starting in the ventronasal retina at ~28 hpf and spreading clockwise across the central retina until it reaches the ventrotemporal side (*Masai et al., 2000*; *Hu and Easter, 1999*;

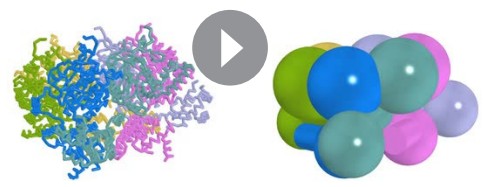

**Video 2.** Time lapse movies of eye vesicle evagination in Z*tcf7l1a*<sup>-/-</sup> mutants. Confocal time lapse movies (1 frame every 5 min) of Z*tcf7l1a*<sup>-/-</sup> mutants (S2, n = 5) expressing the *Tg(rx3:GFP)*<sup>zf460Tg</sup> transgene. First frame taken at 11hpf; membrane RFP in red counterstain.
DOI: https://doi.org/10.7554/eLife.40093.011

*Figure 5A–E,Q*, *Supplementary file 1J*). Although *atoh7* was induced at a similar time in Z*tcf7l1a*<sup>-/-</sup> as in wildtype eyes, the subsequent progression of expression was delayed (*Figure 5F–J,Q*; *Supplementary file 1J*). Classifying the expression of *atoh7* in six categories according to its progression across the neural retina (see legend to *Figure 5*) revealed that *atoh7* expression in mutant retinas was slow to spread and remained restricted to the ventro-nasal or nasal retina for longer (*Figure 5Q*, *Supplementary file 1J*). Indeed, between 36 and 40hpf, Z*tcf7l1a*<sup>-/-</sup> retinas expressed *atoh7* exclusively in the nasal half of the retina (*Figure 5H, Q*), a phenotype we did not see at any stage in sibling embryo eyes. These data indicate that progression of *atoh7* expression and neurogenesis is delayed by about 8–12 hr in Z*tcf7l1a*<sup>-/-</sup> retinas compared to siblings, a timeframe comparable to the delays seen in optic vesicle growth. In line with our results in Z*tcf7l1a*<sup>-/-</sup> embryos, eye vesicle ablated wildtype retinas also showed delayed neurogenesis compared to control non-ablated contralateral eyes at 36hpf (*Figure 5K,L*; 6/6 ablated eyes, two independent experiments).

Our results suggest that retinal precursors in Z*tcf7l1a*<sup>-/-</sup> eyes remain proliferative at stages when precursors in wildtype eyes are already producing neurons.

## Larger eyes undergo premature neurogenesis

Our results are consistent with the idea that neurogenesis may be triggered when the optic vesicle reaches a critical size. To explore this possibility, we generated embryos with larger optic vesicles by overexpressing the Wnt antagonist Dkk1 (*Hashimoto et al., 2000*). Heatshocking *tg(hsp70:dkk1-GFP)*<sup>w32</sup> transgenic embryos at 6hpf led to eyes ~ 34% bigger than control heat-shocked embryos by 28hpf (*Figure 5M,N,R*, n = 12; *Supplementary file 1K*). After 36hpf, wildtype eyes gradually caught up in size as growth slowed in eyes in *dkk1*-overexpressed embryos (*Figure 5R*; *Supplementary file 1K*).

Neurogenesis was prematurely triggered by 28hpf in the eyes of *dkk1* overexpressing embryos, with many more cells expressing *atoh7* compared to eyes in heat-shocked control embryos (*Figure 5M,N*, n = 7 out of 9 embryos). This result is unlikely to be due to a direct effect of *dkk1* overexpression on neurogenesis as premature neurogenesis was not triggered in *tg(hsp70:dkk1-GFP)*<sup>w32</sup> retinas heat-shocked at 24hpf (*Figure 5O,P*, n = 10, 100%). These results further support a link between eye size and the onset of neurogenesis and the size self-regulating ability of the forming eye.

## *tcf7l1a* mutant eyes have more proliferating cells

The delayed production of neurons in *tcf7l1a* mutant eyes suggests that more retina progenotor cells (RPCs) may continue proliferating and contribute to growth compensation. To address this possibility, we counted mitotic phosphohistone3 (PH3) positive cells in wildtype and *tcf7l1a* mutant embryos carrying the *Tg(atoh7:GAP-RFP)*<sup>cu2Tg</sup> transgene that is expressed from the last mitotic event prior to neuronal birth (*Zolessi et al., 2006*).

The proportion of PH3-positive (PH3+) cells standardised to the total number of cells in confocal sections at 36hpf was about 20% higher in Z*tcf7l1a* mutants than wildtypes (*Figure 6A,B,C*, *Supplementary file 1L*, wildtype, n = 7; Z*tcf7l1a*<sup>-/-</sup>, n = 8, p=0.021, unpaired t-test). The proportion of PH3+ cells was similar between nasal and temporal retina and similar between wildtypes and Z*tcf7l1a* mutants (*Figure 6—figure supplement 1*, *Supplementary file 1L*). Furthermore, the percentage of PH3+ RPCs co-expressing *atoh7:GAP-RFP* was 3.6 times higher in wildtypes compared to *tcf7l1a* mutants (*Figure 6A,B,D*, *Supplementary file 1L*, wildtype, 35.21 ± 3.5, n = 7, Z*tcf7l1a*<sup>-/-</sup>, 9.67 ± 1.58, n = 8, unpaired t-test p<0.0001). Wildtype nasal RPCs were about three times more

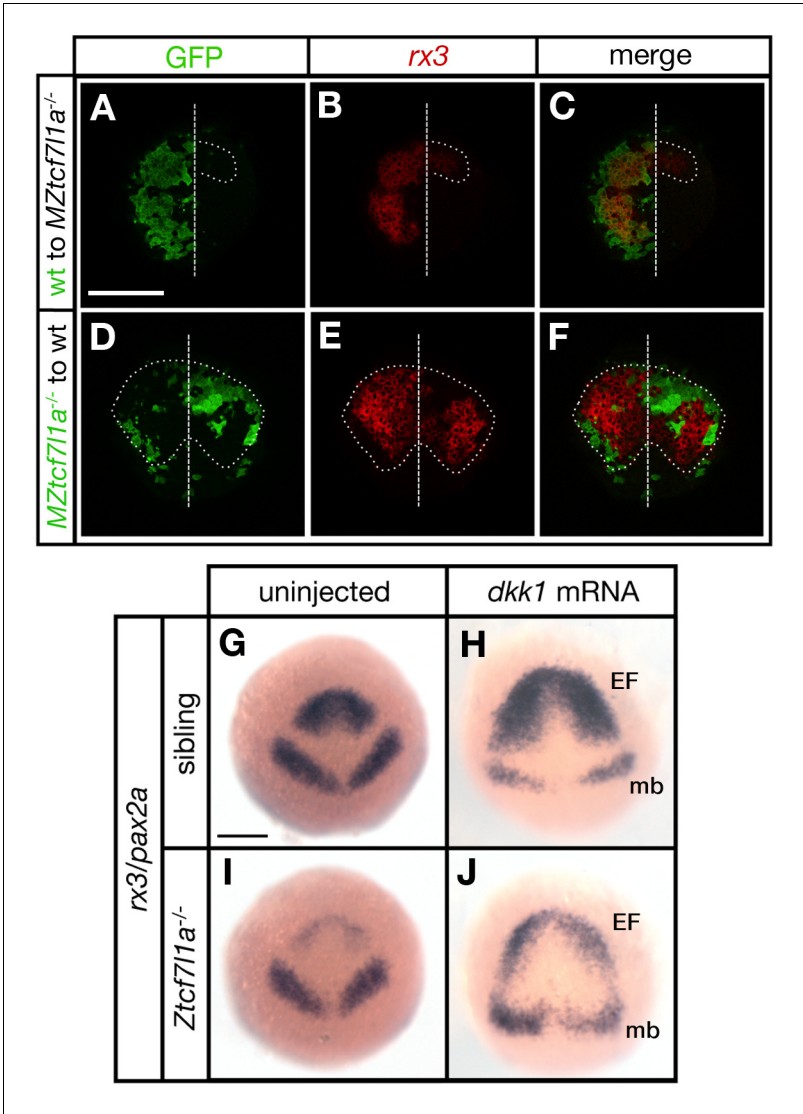

**Figure 3.** Tcf7l1a cell autonomously promotes *rx3* expression in the eye field. (A–F) Dorsal views of confocal images of *rx3* mRNA expression (red) detected by fluorescent *in situ* hybridisation at 10hpf in the anterior neural plates of chimeric embryos containing transplants of (A–C) wildtype (GFP+) donor cells in *MZtcf7l1a*$^{-/-}$ host embryos (100%, n = 13), and (D–F) *MZtcf7l1a*$^{-/-}$ (GFP+) donor cells in wildtype host embryos (100%, n = 9). Dotted line outlines eye fields; note in A-C that *rx3* expression extends considerably caudal to the reduced mutant eye field on the side of the neural plate containing wild-type cells. Dashed line marks the embryo midline. (G–J) *In situ* hybridisation of *rx3* and *pax2a* in sibling (G, H) and *Ztcf7l1a*$^{-/-}$ (I, J) 9hpf embryos, uninjected (G, I) or injected with 50 pg of *dkk1* mRNA (H, J). Abbreviations; EF, eyefield; mb, midbrain Scale Bars = 200 μm.
DOI: https://doi.org/10.7554/eLife.40093.012

likely to be *atoh7:GAP-RFP* positive than nasal RPCs in mutants (**Figure 6D**, **Supplementary file 1L**, wildtype, 45.91 ± 3.71, n = 7, Z*tcf7l1a*$^{-/-}$, 15.44 ± 2.25, n = 8, unpaired t-test p<0.0001), and PH3+-RPCs in the temporal retina of *tcf7l1a* mutants almost never showed *atoh7:GAP-RFP* co-expression (**Figure 6D**, **Supplementary file 1L**, wildtype, 20.97 ± 3.67, n = 7, Z*tcf7l1a*$^{-/-}$, 0.48 ± 0.48, n = 8, unpaired t-test p<0.0001).

These results suggest that at a stage when many wildtype RPCs are undergoing neurogenic divisions, more RPCs in *tcf7l1a* mutants are still proliferating, likely contributing to compensatory growth of the mutant eye.

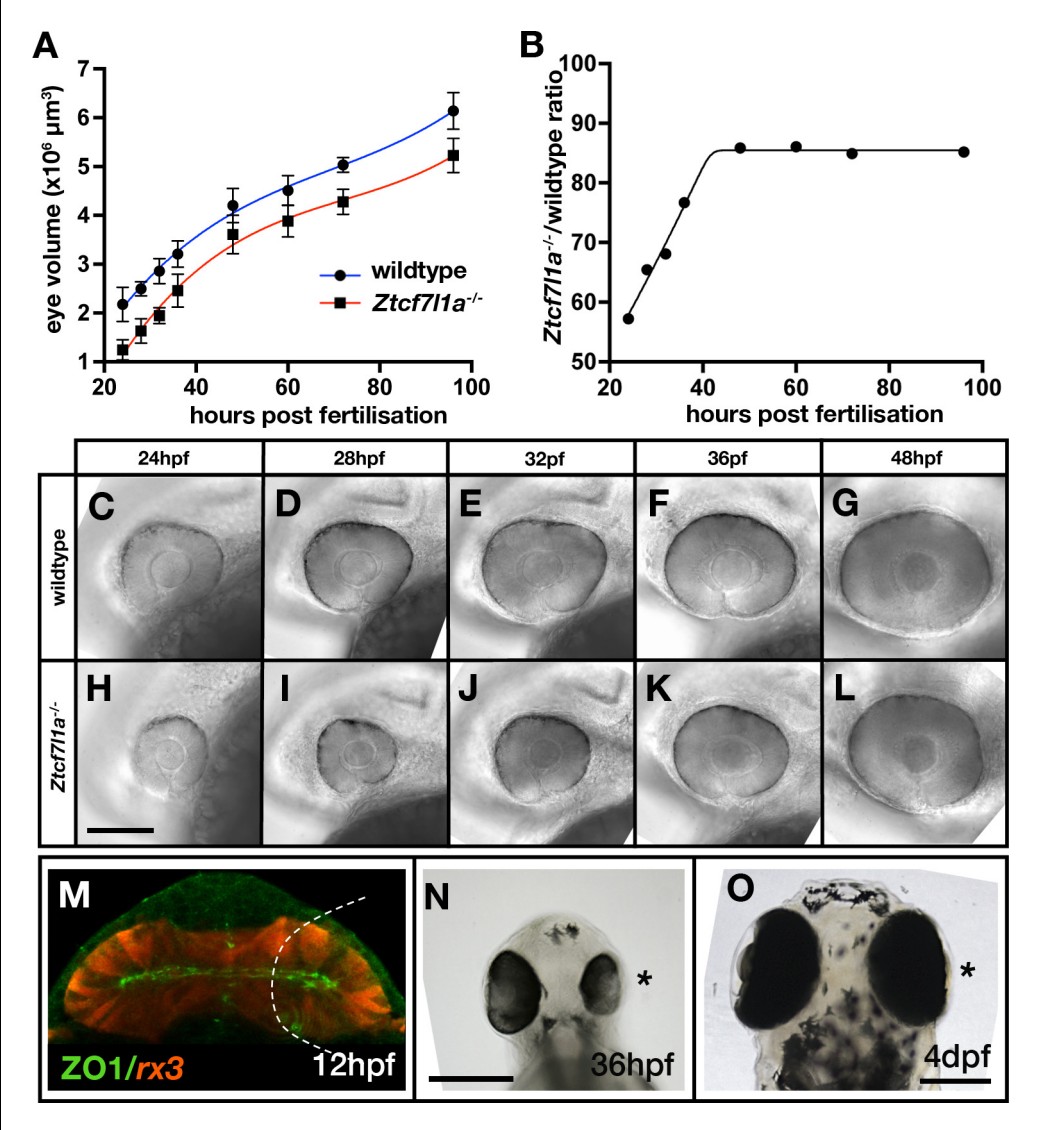

**Figure 4.** Eye size recovers in Ztcf7l1a$^{-/-}$ mutant and eye vesicle-cell removed embryos. (**A**) Growth kinetics of the eye in wildtype (blue line) and Ztcf7l1a$^{-/-}$ (red line) embryos at stages indicated (data in *Supplementary file 1F*, one experiment, 24hpf, wt n = 12, Ztcf7l1a$^{-/-}$ n = 14; 28hpf, wt n = 15, Ztcf7l1a$^{-/-}$ n = 12; 32hpf, wt n = 13, Ztcf7l1a$^{-/-}$ n = 15; 36hpf, wt n = 16, Ztcf7l1a$^{-/-}$ n = 14; 48hpf, wt n = 11, Ztcf7l1a$^{-/-}$ n = 19; 60hpf, wt n = 11, Ztcf7l1a$^{-/-}$ n = 14; 72hpf, wt n = 13, Ztcf7l1a$^{-/-}$ n = 19; 96hpf, wt n = 13, Ztcf7l1a$^{-/-}$ n = 15). (**B**) Plot showing the ratio of Ztcf7l1a$^{-/-}$ to wildtype eye volume from data in (**A**). (**C–L**) Lateral views (dorsal up, anterior to left) of wildtype (**C–G**) and Ztcf7l1a$^{-/-}$ (**H–L**) eyes at stages indicated above panels. (**M–O**) Eye development following partial ablation of the optic vesicle in wildtype embryos at five somite stage. (**M**) Coronal confocal section of evaginating optic vesicles (red) in a wildtype *Tg(rx3:RFP)* five somite stage embryo. Dashed line indicates the approximate extent of ablations performed. 36hpf (**N**) and 4dpf (**O**) eyes in embryos in which cells were unilaterally removed from one optic vesicle (from n = 20). Asterisk indicates the eye that develops from the partially ablated optic vesicle. ZO1, zona ocludens 1. Scale bars = 200 μm.

DOI: https://doi.org/10.7554/eLife.40093.013

The following figure supplements are available for figure 4:

**Figure supplement 1.** Optokinetic response analysis of wildtype and Ztcf7l1a$^{-/-}$ mutants.
DOI: https://doi.org/10.7554/eLife.40093.014

**Figure supplement 2.** Quantification of the posterior lateral line primordium position in wildtype and Ztcf7l1a$^{-/-}$ mutants.
DOI: https://doi.org/10.7554/eLife.40093.015

*Figure 4 continued on next page*

*Figure 4 continued*

**Figure supplement 3.** Cell volume quantification in *tcf7l1a* mutants and siblings.
DOI: https://doi.org/10.7554/eLife.40093.016
**Figure supplement 4.** Growth kinetics of the eye from eye vesicle cell-removed embryos.
DOI: https://doi.org/10.7554/eLife.40093.017

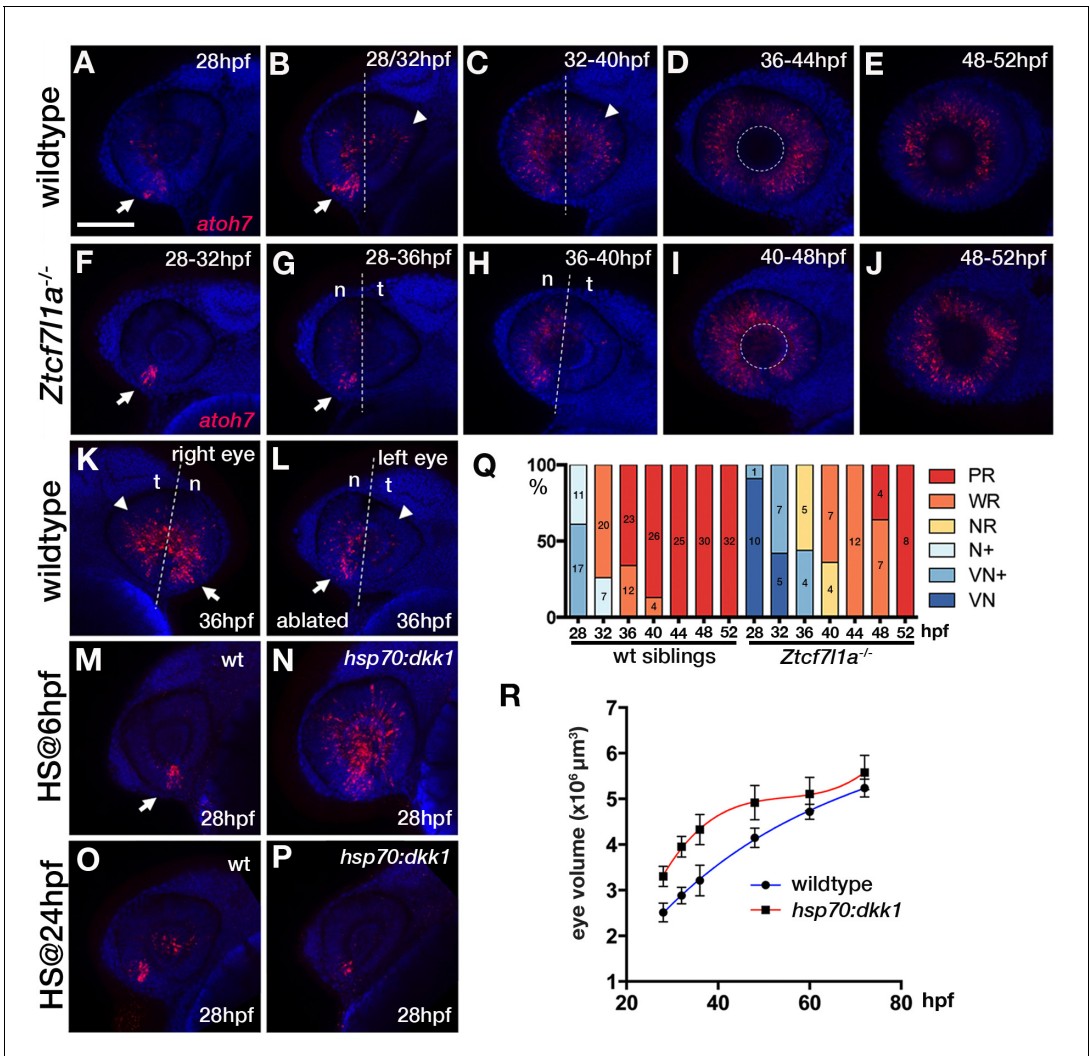

**Figure 5.** Neurogenesis is delayed in small *tcf7l1a*⁻/⁻ eyes and accelerated in large eyes following *hsp70:dkk1* overexpression. (A–P) Lateral views of eyes showing *atoh7* fluorescent *in situ* hybridisation in typical wildtype (A–E, M, O), *Ztcf7l1a*⁻/⁻ (F–J), wildtype left-side optic vesicle-ablated (K, L); from n = 5 embryos) and *Tg(HS:dkk1)*^w32^ (N, P) embryos at stages indicated. (M–P) Wildtype (M, O) and heterozygous sibling *Tg(HS:dkk1)*^w32^ embryos (N, P) heat-shocked at 6hpf (M, N); from n = 7/9 embryos) or 24hpf (O, P); from n = 10/10 embryos) for 45' at 37°C and grown to 28hpf. Anterior is to the left except in (K) in which anterior is to the right. Arrows indicate ventro-nasal retina; arrowheads indicate dorso-temporal retina; dashed line approximate the nasal-temporal division; dashed circle marks lens position. Abbreviations: n, nasal, t, temporal. Scale bar = 100 μm. (Q) Histogram showing the spatial distribution of *atoh7* expression in sibling and *Ztcf7l1a*⁻/⁻ retinas at the indicated hours post-fertilisation (data in **Supplementary file 1F**). VN, ventro nasal expression; VN+, ventro-nasal expression plus a few scattered cells; N+, nasal expression plus scattered cells covering the whole retina; NR, nasal retina expression; WR, whole retina expression; PR, expression localised to the peripheral retina. Numbers in bars represent the number of embryos scored for the particular category of *atoh7* expression. (R) Plot showing the growth kinetics of the eye in wildtype (blue line) and *Tg(HS:dkk1)*^w32^ (red line) embryos at times indicated (data in **Supplementary file 1K**).
DOI: https://doi.org/10.7554/eLife.40093.018

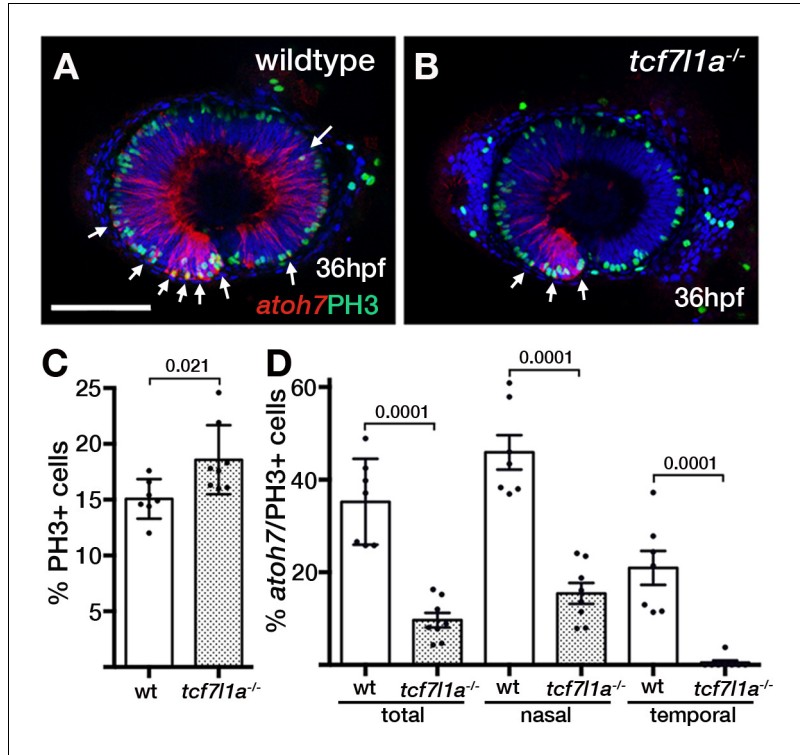

**Figure 6.** Z*tcf7l1a* mutants show more retinal progenitor cells undergoing proliferation. (A–B) Immunostaining detecting phosphohistone3 (PH3, green) and RFP (*Tg(atoh7:GAP-RFP)*$^{cu2Tg-}$, red) in wildtype (**A**) and Z*tcf7l1a*$^{-/-}$ (**B**) eyes at 36hpf . Arrows indicate selected double PH3/RFP positive cells. n, nasal; t, temporal. Scale bar = 100 μm. (**C–D**) Plot showing the percentage of PH3-positive cells (**C**) data in *Supplementary file 1L*) and double PH3/RFP-positive cells (**D**), data in *Supplementary file 1L*). Single experiment, wildtype n = 7, Z*tcf7l1a*$^{-/-}$ n = 8, figures over the bars show p-values from unpaired t-tests.

DOI: https://doi.org/10.7554/eLife.40093.019
The following figure supplement is available for figure 6:

**Figure supplement 1.** Ratio of double atoh7/phosphohistone3 cells in the nasal and temporal retina.
DOI: https://doi.org/10.7554/eLife.40093.020

## An ENU modifier mutagenesis screen in *tcf7l1a* mutant background reveals two groups of genetic modifiers

Although eye formation can recover in *tcf7l1a*$^{-/-}$ mutants despite a much smaller eye field, we speculated that eye development in these embryos might be sensitised to showing the effects of additional mutations. To test this, we performed an ENU mutagenesis screen on fish carrying the *tcf7l1a* mutation (*Figure 7A*).

Homozygous Z*tcf7l1a* mutant adult male fish (F$_0$ founders) were treated with four rounds of ENU (*van Eeden et al., 1999*) and then crossed with Z*tcf7l1a*$^{-/-}$ adult females to generate F$_1$ families (*Figure 7A*). However, possibly because of cellular stress or the synergistic cumulative effect of many mutations induced by ENU, we observed many eyeless F$_1$ embryos. To circumvent this problem, we injected 10 pg/embryo of zebrafish *tcf7l1a* mRNA to rescue any Tcf-dependent eyeless phenotypes in the F$_1$ embryos (*Figure 7A*). Adult F$_1$ fish were outcrossed to *EKW* wildtype fish. All F$_2$ fish were *tcf7l1a*$^{+/-}$ and half carried unknown mutations (m) in heterozygosity (*Figure 7A*). To screen, we randomly crossed F$_2$ pairs from each family aiming for at least 6 clutches of over 100 embryos. The probability of finding double Z*tcf7l1a*$^{-/-}$/m$^{-/-}$ embryos for independently segregating mutations is 1/16, hence we would expect to find ~6 double mutants in 100 embryos. Here, we describe examples of synthetic lethal mutations that lead to microphthalmia/anophthalmia (*U910*; *Figure 7B-K*) or eyes that fail to grow (*U762, U768*; *Figures 8* and *9*).

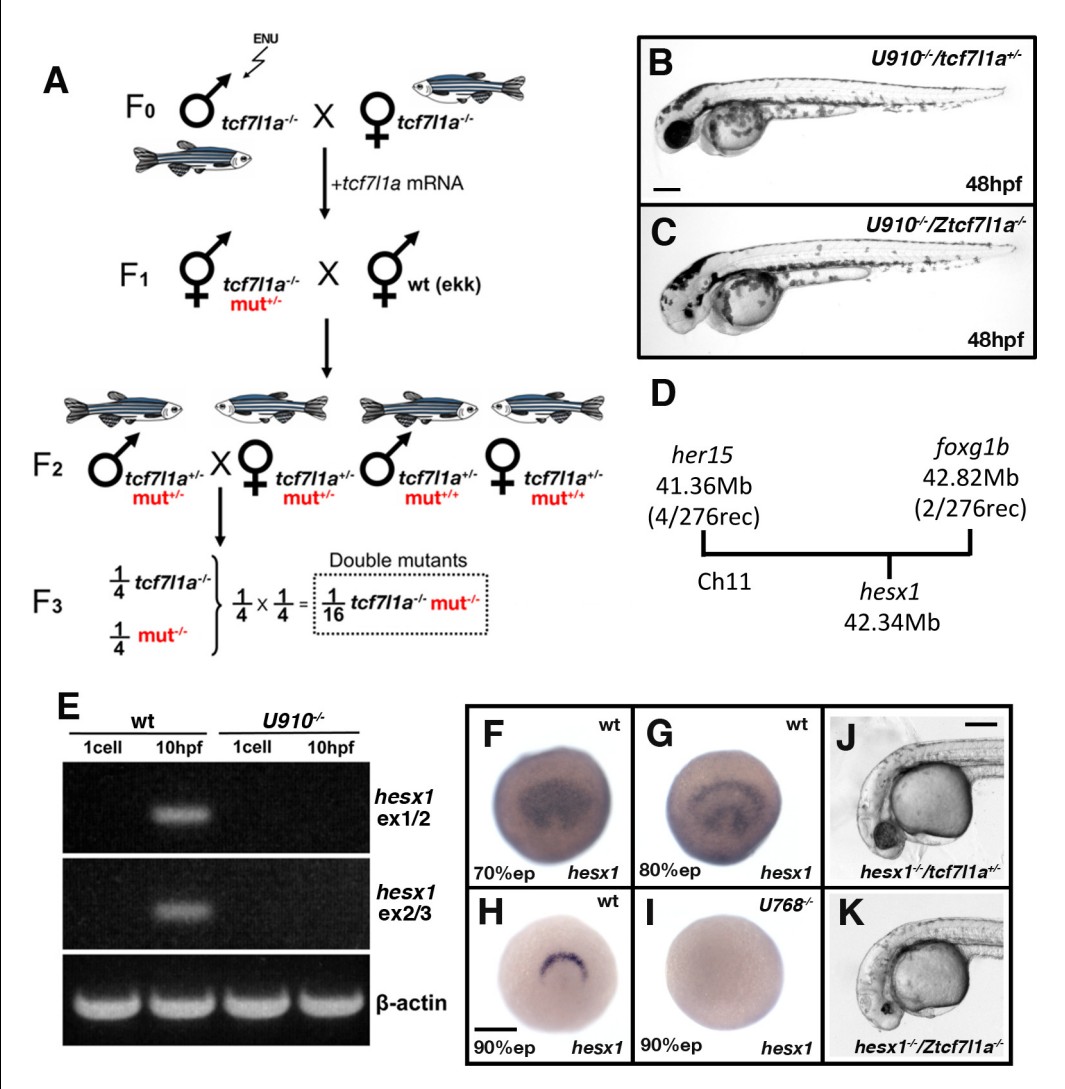

**Figure 7.** Z*tcf7l1a*<sup>-/-</sup> mutants lacking Hesx1 function fail to form eyes. (**A**) Schematic of the genetic strategy to isolate mutations that modify the *tcf7l1a*<sup>-/-</sup> mutant phenotype. (**B–C**) *U910* modifier of the *tcf7l1a*<sup>-/-</sup> mutant phenotype. Lateral views of homozygous *U910* embryos that are heterozygous (**B**) or homozygous (**C**) for the *tcf7l1a* mutation. (**D**) Representation of SSLP segregation linkage analysis mapping of U910 modifier of *tcf7l1a* to a 1.46 megabase (Mb) interval on chromosome 11 (Ch11; rec, recombinants). (**E**) RT-PCR for *hesx1* spanning exons 1–2 (top panel), exons 2–3 (middle panel) and β-actin (bottom panel) on wildtype (lanes 1 and 2) and *U910*<sup>-/-</sup> (lanes 3 and 4) embryo cDNA from 1 cell stage (lanes 1 and 3) and 10hpf (lanes 2 and 4). Single experiment. (**F–I**) *hesx1 in situ* hybridisation on wildtype (**F–H**) and *U910*<sup>-/-</sup> (**I**) embryos at epiboly (ep) stages indicated. Dorsal views, anterior up. (**J, K**) Lateral views of *hesx1*<sup>-/-</sup> (Δex1/2)/*tcf7l1a*<sup>+/-</sup> (**J**) and *hesx1*<sup>-/-</sup> (Δex1/2)/Z*tcf7l1a*<sup>-/-</sup> (**K**) embryos. Four independent experiments, n = 53. Scale bars = 200 μm.

DOI: https://doi.org/10.7554/eLife.40093.021

The following figure supplement is available for figure 7:

**Figure supplement 1.** Genomic DNA sequence of *hesx1* locus spanning exons 1 and 2.

DOI: https://doi.org/10.7554/eLife.40093.022

## A deletion in the *hesx1* locus is a modifier of the *tcf7l1a*<sup>-/-</sup> phenotype that leads to loss of eyes

*tcf7l1a*<sup>-/-</sup> embryos homozygous for the *U910* mutation were eyeless (*Figure 7B,C*) whereas homozygous *U910* mutants with one or no mutant *tcf7l1a* alleles showed no eye phenotype. *U910* was

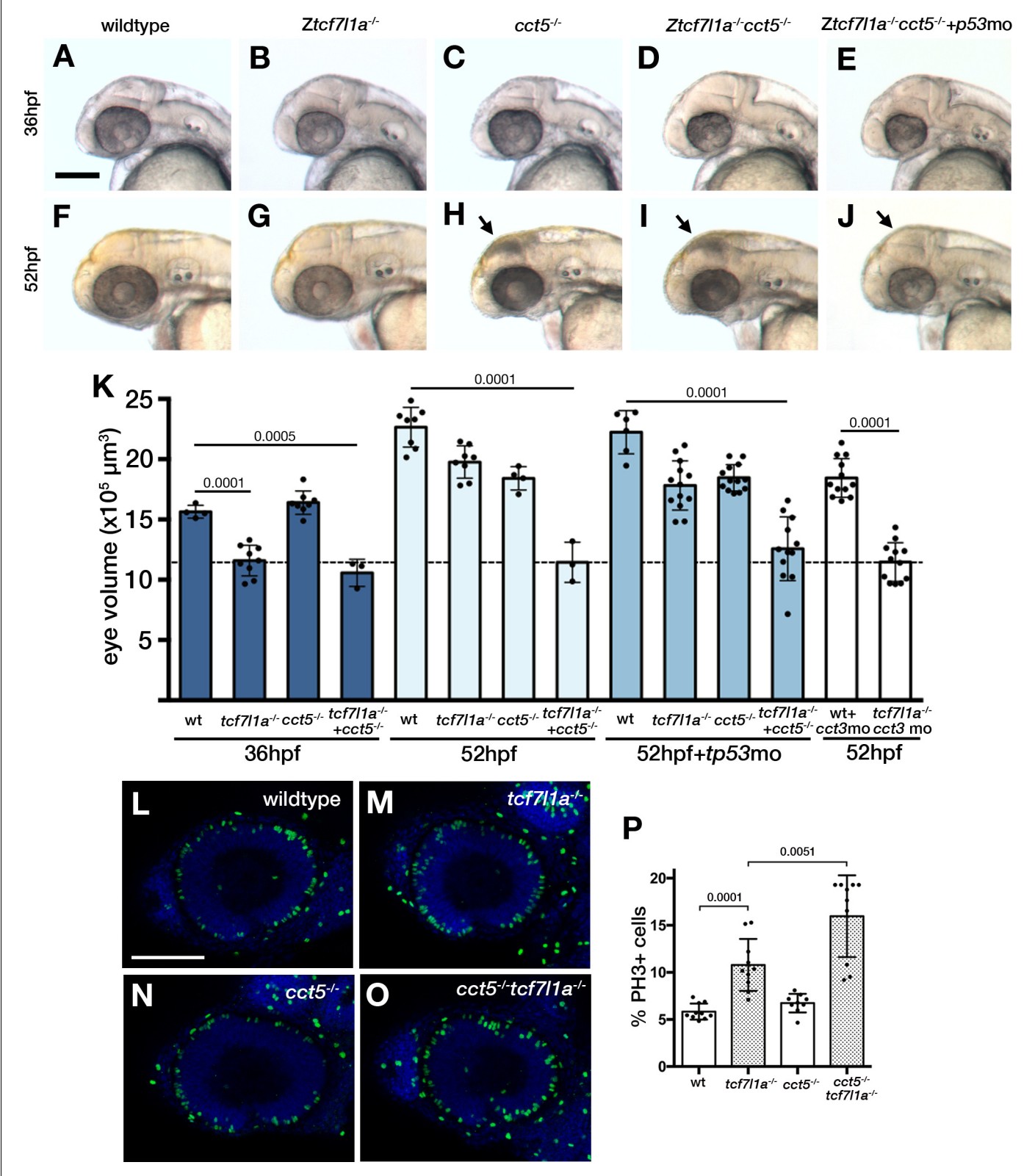

**Figure 8.** Loss of *tcf7l1a* modifies the *cct5^u762* mutant eye phenotype. (A–J) Lateral views of wildtype (A, F), *Ztcf7l1a^-/-* mutant (B, G) *cct5^U762/u762* mutants (C, H), double *cct5^U762/u762*/*Ztcf7l1a^-/-* mutants (D, I) and double *cct5^U762/u762*/*Ztcf7l1a^-/-* mutants injected with 0.8 pmol of *cct3* morpholino (E, J) at indicated stages. Scale bar = 100 μm. Full data in ***Supplementary file 10O***, single experiment, 36hpf, wt n = 4, Z*tcf7l1a^-/-* n = 9, *cct5^-/-* n = 8, *cct5/*

*Figure 8 continued on next page*

*Figure 8 continued*

Ztcf7l1a$^{-/-}$ n = 3; 52hpf, wt n = 8, Ztcf7l1a$^{-/-}$ n = 8, cct5$^{-/-}$ n = 4, cct5/Ztcf7l1a$^{-/-}$ n = 3; 52hpf + 2 pmol *tp53* morpholino, wt n = 6, Ztcf7l1a$^{-/-}$ n = 13, cct5$^{-/-}$ n = 13, cct5/Ztcf7l1a$^{-/-}$ n = 12; 52hpf + 0.8 pmol *cct3* morpholino, wt n = 12, Ztcf7l1a$^{-/-}$ n = 12. (K) Eye volume quantification at the indicated timepoints and conditions shown in A–J) (data in *Supplementary file 1O*). Unpaired t-test. (L–O) Immunostaining detecting phosphohistone3 (PH3, green) in wildtype (L), Ztcf7l1a$^{-/-}$ (M), cct5$^{-/-}$ (N), cct5$^{-/-}$/Ztcf7l1a$^{-/-}$ (O) eyes at 32hpf. (P) Plot showing the percentage of PH3 positive cells in the eyes shown in L–O) (data in *Supplementary file 1Q*) Single experiment, wildtype n = 10, Ztcf7l1a$^{-/-}$ n = 10, cct5$^{-/-}$ n = 9, cct5/Ztcf7l1a$^{-/-}$n = 10, unpaired t-tests.

DOI: https://doi.org/10.7554/eLife.40093.023

The following figure supplements are available for figure 8:

**Figure supplement 1.** Genetic mapping of *U762* and description of the cct5$^{U762}$ mutation.

DOI: https://doi.org/10.7554/eLife.40093.024

**Figure supplement 2.** Whole mount Tunel cell death analysis in *cct5/tcf7l1a* mutants.

DOI: https://doi.org/10.7554/eLife.40093.025

**Figure supplement 3.** *atoh7* expression in *cct5/tcf7l1a* mutants.

DOI: https://doi.org/10.7554/eLife.40093.026

mapped by SSLP segregation analysis (*Kelly et al., 2000*) to a 1.46 Mb interval between 41.36 Mb (four recombinants/276meioses) and 42.82 Mb (two recombinants/276meioses) on chromosome 11 in GRCz10 assembly (*Figure 6D, Supplementary file 1M*). Within this interval is *hesx1*, which morpholino knock-down experiments had previously suggested to genetically interact with *tcf7l1a* (*Andoniadou et al., 2011*). Primers for *hesx1* cDNA failed to amplify in *U910/Ztcf7l1a$^{-/-}$* eyeless embryo cDNA samples. Using a primer set that spans the *hesx1* locus, we found that all *U910/Ztcf7l1a$^{-/-}$* eyeless embryos had a ~ 2700 bp deletion that covers *hesx1* exons 1 and 2 (*hesx1$^{\Delta ex1/2}$*; *Figure 7—figure supplement 1*); this was unexpected as deletions are not normally induced by ENU (see below). Sequencing of the *hesx1* locus revealed that there is a polyA stretch of approximately 80 nucleotides followed by a 33 AT microsatellite repeat on the 3' end of intron two that may have generated a chromosomal instability that led to the deletion of exons 1 and 2 (*Figure 7—figure supplement 1*). As a consequence of the deletion, *hesx1* mRNA was not detected by RT-PCR or in situ hybridisation in *U910* homozygous embryos (*Figure 6E,H,I*). We further confirmed that only *U910*-F$_2$ embryos that were homozygous for both the *tcf7l1a* mutation and *hesx1$^{U910/U910}$* were eyeless (*Figure 7B,C*).

As ENU usually generates point mutations, we speculated that the deletion in *hesx1$^{\Delta ex1/2}$* was not caused by our mutagenesis but was already present in one or more fish used to generate the mutant lines. Indeed, we found the same deletion in wildtype fish not used in the mutagenesis project. To confirm that the eyeless phenotype in *hesx1$^{U910/U910}$/Ztcf7l1a$^{-/-}$* double mutants is not caused by another mutation induced by ENU, we crossed *Ztcf7l1a$^{-/-}$* fish to one such wildtype *TL* fish carrying *hesx1$^{\Delta ex1/2}$*. Incrossing of *hesx1$^{\Delta ex1/2/\Delta ex1/2}$/tcf7l1a$^{+/-}$* adult fish led to embryos with a very small rudiment of eye pigment with no detectable lens (*Figure 7J,K*). Genotyping of eyeless and sibling embryos confirmed that only double homozygosity for *hesx1$^{\Delta ex1/2}$/Ztcf7l1a$^{-/-}$* led to the eyeless embryo phenotype (Four independent experiments, n = 53, *Supplementary file 1N*).

The interaction between *hesx1* and *tcf7l1a* mutations strikingly illustrates how the developing eye can fully cope with loss of function of either gene alone but fails to form in absence of both gene activities. Additional eyeless families that do not carry the *hesx1* deletion were identified but they remain to be validated and mutations cloned.

### *cct5* and *gdf6a* mutations compromise the ability of *tcf7l1a* mutant eyes to undergo compensatory growth

*U762* mutant eyes showed no significant size difference compared to wildtypes at 36hpf (*Figure 8A, C,K, Supplementary file 1O*); neither did *Ztcf7l1a$^{-/-}$* compared to double *U762/Ztcf7l1a$^{-/-}$* eyes (*Figure 8B,D,K, Supplementary file 1O*). However, by 52hpf *U762* mutants showed slightly reduced eye size and this phenotype was considerably more severe in embryos additionally homozygous for the *Ztcf7l1a* mutation (*Figure 8F–I,K, Supplementary file 1O*).

The *U762* mutation was mapped by SSLP segregation analysis to a 1.69 Mb interval between 15.50 Mb and 17.19 Mb on chromosome 24 (*Figure 8—figure supplement 1A*) and through sequencing candidate genes in this interval (*Figure 8—figure supplement 1A; Supplementary file 1P*), we identified a mutation in the splice donor of *cct5* (chaperonin containing

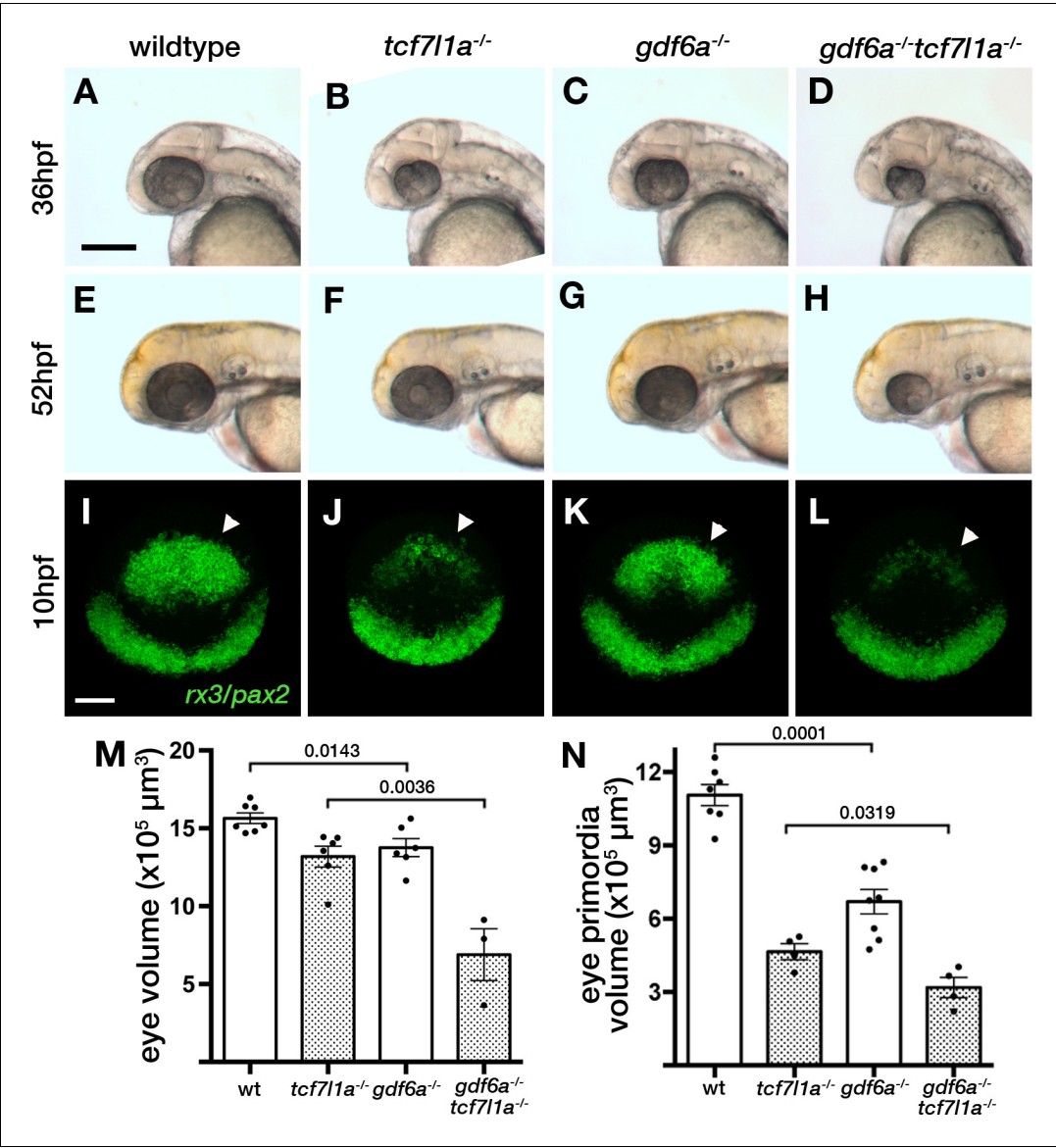

**Figure 9.** Loss of *tcf7l1a* modifies the *gdf6a*^U768/U768^ mutant eye phenotype. (A–H) Lateral views of eyes in wildtype (A, E), *Ztcf7l1a*^-/-^ (B, F), *gdf6a*^U768/U768^ (C, G) and double *gdf6a*^U768/U768^/*Ztcf7l1a*^-/-^ (D, H) embryos at 36hpf (A–D) and 52hpf (E–H). Dorsal up, anterior to left. Arrows indicate the lens. Scale bar = 200 μm. (I) Whole mount fluorescent in situ hybridisation for *rx3* and *pax2a* in wildtype (I), *Ztcf7l1a*^-/-^ (J), *gdf6a*^U768/U768^(K) and double *gdf6a*^U768/U768^/*Ztcf7l1a*^-/-^ (L) embryos at 10hpf. Dorsal view, anterior up. Arrows point to *rx3* eye field expression. Scale bar = 100 μm. (M) Eye volume quantification in wildtype (n = 7), *Ztcf7l1a*^-/-^ (n = 6), *gdf6a*^-/-^(n = 6) and *gdf6a*^U768/U768^/*Ztcf7l1a*^-/-^ double mutant siblings (n = 3) at 36hpf (data in ***Supplementary file 1R***). Single experiment, unpaired t-test. (N) Eye field volume quantification from *rx3* fluorescent *in situ* hybridisation shown in I-L in wildtype (n = 7), *Ztcf7l1a*^-/-^ (n = 4), *gdf6a*^-/-^(n = 8) and *gdf6a*^U768/U768^/*Ztcf7l1a*^-/-^ double mutant siblings (n = 4) at 10hpf (data in ***Supplementary file 1S***). Single experiment, unpaired t-test.
DOI: https://doi.org/10.7554/eLife.40093.027

TCP-1 epsilon) intron 4 (GT >GC, ***Figure 8—figure supplement 1B***). The mutation leads to the usage of an alternative splice donor in the 3' most end of *cct5* exon 4, which induces a two nucleotide deletion in the mRNA (***Figure 8—figure supplement 1C***). This deletion changes the reading frame of the protein after amino acid 176, encoding a 29aa nonsense stretch followed by a stop codon (***Figure 8—figure supplement 1C,D***). The mutation also induces nonsense-mediated decay of the mRNA (not shown). *U762* and *cct5*^hi2972bTg^ mutations failed to complement (not shown)

supporting the conclusion that the *cct5* mutation in *U762* is responsible for the *tcf7l1a* modifier phenotype. Cct5 is one of the eight subunits of the chaperonin TRiC/TCP-1 protein chaperone complex, which assists the folding of actin, tubulin and many proteins involved in cell cycle regulation (*Sternlicht et al., 1993*; *Dekker et al., 2008*; *Yam et al., 2008*).

To assess if the phenotype in double *cct5^U762/tcf7l1a* mutants is likely due to TRiC/TCP-1 chaperone activity or an independent function of Cct5 we knocked down *cct3*, another member of the chaperonin complex, in *tcf7l1a* mutants. Morpholino knockdown of *cct3* abrogated eye growth in Z*tcf7l1a* mutants as in *cct5^U762*/Z*tcf7l1a* mutants (*Figure 8K*, last two bars; *Supplementary file 1O*), suggesting that the genetic interaction is between TRiC/TCP-1 and Tcf7l1a function.

Compared to single *cct5^U762* or *tcf7l1a* mutants, double *cct5^U762*/Z*tcf7l1a* homozygous mutant eyes did not grow beyond 36hpf (*Figure 8* compare D, I to B. G, *Figure 8K*, dotted line, *Supplementary file 1O*). As described for other *cct* gene mutants (*Matsuda and Mishina, 2004*), we observed dying cells in 36hpf and 48hpf *cct5^U762*and *cct5^U762*/Z*tcf7l1a* mutant eyes and tecta (*Figure 8—figure supplement 2C*), whereas dying cells were rarely detected in these regions at these times in wildtype or Z*tcf7l1a^-/-* mutant siblings (*Figure 8—figure supplement 2*). To assess if apoptosis contributes to the lack of compensatory eye growth in *cct5^U762*/Z*tcf7l1a* mutants, we inhibited cell death by knocking down *tp53* (*Figure 8E,J,K*, *Supplementary file 1O*). Double *cct5^U762*/Z*tcf7l1a* mutants with abrogated Tp53 function showed little or no apoptosis in the eye or tectum (*Figure 8*, compare J to I, arrows; *Figure 8—figure supplement 2E,J,O*) but still showed eye size reduced similarly to *cct5*/Z*tcf7l1a* mutants at 36hpf and 52hpf (*Figure 8E,J,K*; *Supplementary file 1O*).

As Cct5 is implicated in the folding of cell cycle related proteins, we assessed the presence of proliferative RPCs and neurons in *cct5^U762*/Z*tcf7l1a* mutants (*Figure 8L–P*, *Supplementary file 1Q*). *cct5^U762* mutants showed no significant difference in PH3 +cells compared to wildtype siblings (*Figure 8L,N,P*, *Supplementary file 1Q*, n = 9), whereas double *cct5^U762*/Z*tcf7l1a^-/-* mutants showed a 48% increase compared to single Z*tcf7l1a* mutants (*Figure 8M,O,P*, *Supplementary file 1Q*, n = 10, p<0.0051, unpaired t-test, two experiments). By 48hpf, wildtype eyes show strong *atoh7*: GFP expression in the retinal ganglion cell layer (*Figure 8—figure supplement 3A*, n = 4, arrow head; *Masai et al., 2003*), whereas although *cct5^U762/U762* eye cells do express GFP, lamination of neurons is abnormal (*Figure 8—figure supplement 3B*, n = 5). In *cct5^U762/U762*/Z*tcf7l1a^-/-* eyes, GFP-expressing neurons are almost completely restricted to the ventro-nasal retina (*Figure 8—figure supplement 3C*, n = 6).

Homozygous *U768* mutants show slightly smaller (*Figure 9A,C,M*, *Supplementary file 1R*, n = 6, p=0.0143, unpaired t-test) and misshapen eyes; this mutation was mapped to *gdf6a* (*Valdivia et al., 2016*). Eyes in *gdf6a^U768/U768*/Z*tcf7l1a^-/-* embryos at 36hpf were reduced to 52% the size of eyes in Z*tcf7l1a^-/-* mutants (*Figure 9B,D,M*, *Supplementary file 1R*, n = 3, p=0.0036, unpaired t-test). Unlike Z*tcf7l1a^-/-* mutants in which eye size recovered, eyes in *gdf6a^U768/U768*/Z*tcf7l1a^-/-* embryos remained smaller than in single mutants or wildtypes at 52hpf (*Figure 9E–H*). This suggests that the ability to compensate eye size is compromised in absence of both *gdf6a* and *tcf7l1a* function.

The smaller eye in *gdf6a^U768/U768*/Z*tcf7l1a^-/-* mutants at 36hpf suggested that early eye development and maybe eye field specification in these mutants is compromised. Volumetric analysis of *rx3* expression by fluorescent *in situ* hybridisation showed that *gdf6a^-/-* eye fields were reduced to 63% of wildtype size at 10hpf (*Figure 9I,K,N*, arrowheads; *Supplementary file 1S*, n = 8, p=0.0001, unpaired t-test). Moreover, *gdf6a^U768/U768*/Z*tcf7l1a^-/-* eye fields were about ~68% of the size of Z*tcf7l1a^-/-* mutants (*Figure 9I–L,N*, arrowheads; *Supplementary file 1S*, n = 4, p=0.0143, unpaired t-test).

Altogether, analysis of the interacting mutations reveals that although abrogation of Tcf7l1a function alone has little effect on formation of eyes, it can lead to complete loss of eye formation or more severe eye phenotypes in combination with additional mutations. Consequently, although eye development is sufficiently robust to cope with loss of Tcf7l1a, mutant embryos are sensitised to the effects of additional mutations.

## Discussion

In this study, we show that although Tcf7l1a is required for cells to adopt eye field identity and express *rx3*, *tcf7l1a* mutants form normal, functional eyes. This finding reveals a remarkable ability of

the eye to develop normally from an eye field that is half the size of that in the wildtype condition. Tcf function in *tcf7l1a* mutants is not genetically compensated by upregulation of other *tcf* genes nor by other genetic mechanisms that restore neural plate regionalisation and eye field formation. Instead, we find that *tcf7l1a* mutant optic vesicles maintain more proliferative RPCs and delay neurogenesis enabling size recovery. We observe a similar effect when optic vesicle cells are physically ablated. In contrast, neurogenesis is prematurely induced in larger optic vesicles, likely depleting progenitors and slowing growth. Our results suggest that size-dependent regulation of the balance between proliferation and differentiation may buffer the developing eye against initial differences in cell number. Although the developing eye can cope with loss of Tcf7l1a function, we speculated that embryos lacking Tcf7l1a would not be robust to the consequences of additional mutations affecting eye formation. In support of this, we identify mutations in three other genes that give synthetically enhanced eye phenotypes when combined with the *tcf7l1a* mutation. This approach facilitates identification of genes that participate in genetic networks that make developing eyes robust to mutations that compromise eye field specification and optic vesicle growth.

## The *tcf7l1a* mutation is fully penetrant with no apparent genetic compensation during neural plate patterning

*Tcf7l1* is a core Wnt/β-catenin pathway transcription factor that can activate or repress genes dependent upon the status of the Wnt signalling cascade (*Cadigan and Waterman, 2012*). Homozygous *tcf7l1* mutant mice present severe mesodermal and ectodermal patterning defects (*Merrill et al., 2004*), but the duplication of *tcf7l1* into *tcf7l1a* and *tcf7l1b* in zebrafish has led to functional redundancy (*Dorsky et al., 2003*).

Although Z*tcf7l1a* embryos have a severe eye field specification phenotype, they still develop normal eyes. We confirmed that the *tcf7l1a^{m881}* mutant allele is null, generates no wildtype transcript and that morpholino knock-down specifically of Tcf7l1a does not give an eyeless phenotype. Hence, the originally described MZ*tcf7l1a* eyeless phenotype (*Kim et al., 2000*) may have been due to genetic background effects modifying the outcome of the *tcf7l1a^{m881}* allele. The fact that we were able to recover an eyeless modifier of the *tcf7l1a* phenotype in our own mutagenesis pilot screen lends support to this idea.

At the stage of eye specification, we did not find genetic compensation in *tcf7l1a* mutants by other *tcf* genes. Even though *tcf7l1a* mutants develop eyes, they do so from an eye field that is ~50% smaller than wild-type. Although we did not find evidence for genetic compensation, and despite *tcf7l1* being duplicated in fish, the fact that neither gene has been lost due to genetic drift suggests that having both genes may confer enhanced fitness and robustness to zebrafish. As an example, paralogous Lefty proteins make Nodal signalling more stable to noise and perturbations during early embryogenesis (*Rogers et al., 2017*).

Tcf7l1a is cell-autonomously required for the expression of *rx3* and consequently is a bona fide eye field gene regulatory network transcription factor that functions upstream to *rx3*. *tcf7l1a* is expressed very early in the anterior neural plate and so may work alongside *otx*, *sox*, *six* and *pax* genes to regionalise the eye-forming region of the neural plate (*Beccari et al., 2013*; *Zuber et al., 2003*). Considering that it is the repressor activity of Tcf7l1a that promotes eye formation (*Kim et al., 2000*), the most likely role for Tcf7l1a is to repress transcription of a gene that suppresses eye field formation.

## Compensatory tissue growth confers robustness to eye development

We show that despite the small eye field in *tcf7l1a* mutants, the optic vesicles evaginate and undergo overtly normal morphogenesis. Although *tcf7l1a* mutant eye vesicles are still much smaller than wild-type at 24hpf, we found that their eye growth kinetics and cell volumes are similar. This suggests that the mechanisms that regulate overall growth of the retina in both conditions are comparable albeit delayed in the *tcf7l1a* mutant retina.

Although *atoh7* expression is initiated in the ventronasal retina in *tcf7l1a* mutants at the same stage as in wild-type eyes, the wave of *atoh7* expression that spreads across the retina is delayed by approximately 8–12 hr in mutants. *atoh7* is required for the first wave of neurogenesis in the retina (*Brown et al., 2001*; *Kay et al., 2001*; *Wang et al., 2001a*) and thus, the delay we see in *tcf7l1a* mutants suggests that RPCs continue proliferating in mutants at stages when they are already

generating neurons in wild-type eyes. Indeed, we show that the *tcf7l1a* mutant eye has more mitotic RPCs, fewer of which are undergoing neurogenic divisions. This suggests that the extended period of proliferative growth due to delayed neurogenesis enables the forming eye to continue growing and recover its size. We observed a similar phenomenon of delayed neurogenesis and prolonged growth when cells were removed from one optic vesicle. Conversely, *atoh7* spreads precociously in experimentally enlarged optic vesicles. The premature neurogenesis of RPCs in these conditions may contribute to eyes achieving a final size similar to wild-type. Altogether, our data suggest that the timing of the spread of neurogenesis across the retina may be coupled to size of the eye, thereby providing a mechanism to buffer eye size. It is intriguing that the compensatory changes in growth seen in *tcf7l1a* mutant and optic-vesicle ablated eyes seem to occur prior to the establishment of the ciliary marginal zone, which accounts for the vast majority of eye growth (*Fischer et al., 2013*).

Our results support classical embryology experiments from Ross Harrison, Victor Twitty and others (*Harrison, 1929*; *Twitty and Schwind, 1931*; *Twitty and Elliott, 1934*). These investigators showed that when eye primordia from small-eyed salamander species (*A. punctatum*) were transplanted to larger-eyed salamanders (*A. tigrinum*) or vice-versa, the eye derived from the grafted tissue formed an eye of a size corresponding to the donor salamander species. Species-specific size differences are also observed in self-organising in vitro cultured eye organoids derived from mouse or human embryonic stem cells (*Nakano et al., 2012*). Our work, together with the experiments in salamanders and organoids, suggests that the developing eye has intrinsic size-determining mechanisms.

Size regulatory mechanisms have been previously described in other species and perhaps most extensively studied in the fly wing imaginal disc (*Potter and Xu, 2001*). Indeed, many models have been put forward to explain imaginal disk size control (*Eder et al., 2017*; *Irvine and Shraiman, 2017*; *Vollmer et al., 2017*). It is evident that the final size of paired structures within individuals is remarkably similar supporting the idea that the mechanisms that control the size of such organs/tissues are highly robust.

## Addressing the robustness of eye formation through a forward mutagenesis screen in fish carrying the *tcf7l1a* mutation

Our results indicate that *tcf7l1a* mutant eyes are sensitised to the effects of additional mutations. Indeed, a homozygous deletion of the two first exons of *hesx1* leads to eyeless embryos when in combination with *tcf7l1a*. This result also confirms our previous observations suggesting a genetic interaction between *hesx1* and *tcf7l1a* based upon morpholino knock-down experiments (*Andoniadou et al., 2007*). Furthermore, both *hesx1* and *tcf7l1a* are expressed in the anterior neural plate including the eye field, and as observed in *tcf7l1a* zebrafish mutants, *hesx1* mutant mice also show a posteriorised forebrain (*Andoniadou et al., 2007*; *Martinez-Barbera et al., 2000*). These and our results suggest that Tcf7l1a and Hesx1 have similar, overlapping functions in the anterior neural plate such that the eyeless phenotype is expressed in zebrafish only when both genes are abrogated. Mutations in *hesx1* lead to anophthalmia, microphthalmia, septo-optic dysplasia (SOD) and pituitary defects in humans and mice (*Dattani et al., 1998*; *Gaston-Massuet et al., 2008*; *Martinez-Barbera et al., 2000*; *Thomas et al., 2001*). Interaction of *hesx1* mutations with other genetic lesions may also occur in patients carrying Hesx1 mutations, as the phenotypes in these individuals show variable expressivity (*McCabe et al., 2011*). In these patients, *tcf7l1a* should be considered as a candidate modifier for *hesx1*-related genetic conditions.

*Gdf6a* is a TGFβ pathway member (*David and Massagué, 2018*) that when mutated in zebrafish results in small mis-patterned eyes, neurogenesis defects and retino-tectal axonal projection errors (*Gosse and Baier, 2009*; *French et al., 2009*). In humans, mutations in *GDF6* have been identified in anophthalmic, microphthalmic and colobomatous patients (*Asai-Coakwell et al., 2009*) as well as in some cases of Leber congenital Amaerurosis (*Asai-Coakwell et al., 2013*). Double *gdf6a*[U768]/*tcf7l1a* mutant eye fields are smaller than both single mutants and their eyes fail to recover their size at later stages. This suggests that *gdf6a/tcf7l1a* double mutant eye fields have more severe specification defects compared to either individual mutant, and that double mutant optic vesicles lack the compensatory growth seen in *tcf7l1a* mutants. Given the phenotypes we describe, it is perhaps surprising that *gdf6a* appears not to be expressed in the eye field (*Rissi et al., 1995*), although it is expressed in neighbouring tissues and also prior to eye specification (*Sidi et al., 2003*).

Consequently, we presume that Gdf6 acting prior to eye field formation or arising from outside of the eye field impacts eye field specification. It is also intriguing that *gdf6a* mutants show premature expression of *atoh7* and neurogenesis (*Valdivia et al., 2016*). If this phenotype is epistatic to the compensatory growth mechanisms, then this may contribute to the lack of growth in double mutant eyes.

Mutations in *cct5* in combination with *tcf7l1a* also led to phenotypes in which eye size failed to recover. *cct5* codes for the epsilon subunit of the TCP-1 Ring Complex (TRiC) chaperonin that is composed of eight different subunits that form a ring, the final complex organised as a stacked ring in a barrel conformation (*Yébenes et al., 2011*). In vitro studies indicate TRiC chaperonin mediates actin and tubulin folding (*Sternlicht et al., 1993*); however, it also assists in the folding of cell cycle-related and other proteins (*Dekker et al., 2008*; *Yam et al., 2008*). A mutation in *cct2* has been found in a family with Leber congenital amaurosis retinal phenotype (*Minegishi et al., 2016*; *Minegishi et al., 2018*) and mutations in *cct4* and *cct5* have been related to sensory neuropathy (*Pereira et al., 2017*; *Lee et al., 2003*; *Hsu et al., 2004*; *Bouhouche et al., 2006*). Similar to our *cct5* mutant, *cct1, cct2, cct3, cct4 and cct8* mutant zebrafish show retinal degeneration (*Berger et al., 2018*; *Matsuda and Mishina, 2004*; *Minegishi et al., 2018*), suggesting that the *cct5/tcf7l1a* double mutant phenotype is due to abrogation of TRiC chaperonin function, a conclusion supported by *cct3* knockdown in *tcf7l1a* mutants. Double *cct5/tcf7l1a* homozygous mutant eyes degenerate prematurely and to a greater extent than *cct5* single mutants, and neurogenesis is also severely compromised. However, the lack of compensatory growth is not solely due to cell death as blocking apoptosis in *cct5/tcf7l1a* mutants fails to restore eye size. Our results show that the consequence of *cct5* loss of function is exacerbated by the lack of *tcf7l1a* function, although it is currently unclear how such an interaction might occur. However, this genetic interaction does highlight that in some conditions a gene of pleiotropic function, like *cct5,* can lead to a specific phenotype in the eye.

Surprisingly many eyeless embryos were observed in the F1 clutches of embryos used to establish homozygous *tcf7l1a* fish carrying new mutations. These phenotypes were suppressed by providing exogenous wildtype *tcf7l1a*. It is not unusual to see mutant phenotypes in F1 embryos in ENU screens and we suspect that heavy mutational load may impact developmental processes such that phenotypic penetrance is enhanced due to cellular or tissue level stress. Additionally, we now think it likely that the *hesx1* mutant allele was in the background of some parent fish and this may also have contributed to enhancing the homozygous *tcf7l1a* phenotype in combination with the many newly induced ENU mutations.

Anophthalmia and microphthalmia are generally associated with eye field specification defects (*Reis and Semina, 2015*), but given that normal eyes can still develop from a much reduced eye field, further analysis of the genetic and developmental mechanisms that lead to small or absent eyes is warranted. Our isolation and identification of modifiers of *tcf7l1a* highlights the utility of genetic modifier screens to identify candidate genes underlying congenital abnormalities of eye formation. Indeed, given that Tcf7l1a itself can now be classified as a *bona fide* gene in the eye transcription factor regulatory network, it should be considered when screening patients with inherited morphological defects in eye formation.

## Materials and methods

### Animal use, mutant and transgene alleles, genotyping and heat shock

Adult zebrafish were kept under standard husbandry conditions and embryos were obtained by natural spawning. Wildtype and mutant embryos were raised at 28.5°C and staged according to *Kimmel et al. (1995)*. To minimise variations in staging, embryos were collected every 30 min and kept separate clutches according to their time of fertilisation. Fish lines used were *tcf7l1a/headless (hdl)*[m881] (*Kim et al., 2000*), *cct5*[hi2972bTg] (*Amsterdam et al., 2004*), *cct5*[U762], *gdf6a*[U768] (*Valdivia et al., 2016*), *hesx1*[U910], *tcf7l1b*[zf157Tg] (*Gribble et al., 2009*), *Tg(atoh7:GFP)*[rw021Tg] (*Masai et al., 2000*), *Tg(atoh7:GAP-RFP)*[cu2Tg] (*Zolessi et al., 2006*), *Tg(hsp70:dkk1-GFP)*[w32] (*Stoick-Cooper et al., 2007*) and *Tg(rx3:GFP)*[zf460Tg] (*Brown et al., 2010*). All the alleles except for *cct5*[hi2972bTg] were genotyped by KASP assays (K Biosciences, assay barcodes: 1077647141 (*cct5*[U762]), 1077647146 (*gdf6a*[U768]), 172195883 (this assay discriminates a SSLP 500 bp from the 3'

end of the deletion in *hesx1*[U910]), 1145062619 (*tcf7l1a*[m881])) using 1 µl of genomic DNA for 8 µl of reaction volume PCR as described by K Biosciences.

For heatshock (HS) gene induction, embryos from a heterozygous *Tg(hsp70:dkk1-GFP)*[w32] to wild type cross were moved from embryo media at 28.5°C to 37°C at 6hpf or 24hpf for 45 min, and then back to 28.5°C embryo media. Three hours post HS, embryos were separated in controls (GFP-) and HS experimental (GFP+) groups, and fixed at the stages described in results.

## ENU mutagenesis and mutant mapping

Homozygous male *tcf7l1a*[m881] fish were exposed to four rounds of ENU according to *van Eeden et al. (1999)*. Details of the mutagenesis pipeline are in the results section. Embryos from incrosses of carriers of the *cct5*[U762] or *gdf6a*[U768] mutations, which show a phenotype as homozygous embryos independently of mutations in *tcf7l1a*, were identified for the described eye phenotype at 3dpf to avoid ambiguity and false positives. For rough mapping, batches of 30 mutants and 30 siblings were fixed in methanol and genomic DNA was extracted by proteinase K protocol. This gDNA was then used for bulk segregant analysis PCR to test a library of 245 polymorphic SSLP variants spanning the whole zebrafish genome (*Stickney et al., 2002*). SSLP markers heterozygous in the sibling samples and homozygous in the mutant sample were confirmed on gDNA samples of 12 mutant and 12 sibling individuals. Markers that showed linkage to a locus were tested on additional mutant samples, and more SSLP markers were tested for the mapped region until a genomic interval was defined.

Homozygous *tcf7l1a/hesx1*[U910] mutant carriers were incrossed, and eyeless embryos and siblings were fixed in methanol. Rough mapping was carried out as above but in this case sibling embryos used for bulk segregant analysis were genotyped for *tcf7l1a*[m881] and only homozygous mutants with eyes were included in the sibling pool.

## mRNA synthesis, embryo microinjection and morpholinos

mRNA for overexpression was synthesised using RNA mMessage mMachine transcription kits (Ambion). One- to two-cell stage embryos were co-injected with 10 nl of 5 pg of GFP mRNA and morpholinos or in vitro synthesised mRNA at the indicated concentrations. Only embryos with an even distribution of GFP fluorescence were used for experiments.

For cell volume analysis, one-cell stage embryos from a *tcf7l1a*$^{\pm}$ incross were injected with 5 pg pCS2-GFP DNA and 10 pg lyn-cherry mRNA. The following day, embryos were sorted for GFP mosaicism in the eye and mounted in PTU/Tricaine-containing 1% low-melt agarose and were imaged at 24 and 36hpf in a Leica SP8 confocal microscope.

Morpholino sequences: mo2 t*cf7l1a* (5' AGG CAT GTT GGC ACT TTA AAT G 3'), mo[*tcf7l1b*] (5'-CAT GTT TAA CGT TAC GGG CTT GTC T-3'; *Dorsky et al., 2003*) and mo[C] (TGT TGA AAT CAG CGT GTT CAA G). *tcf7l1a*[m881/m881] embryos injected with mo[*tcf7l1b*] phenocopy the loss of eye phenotype seen in *tcf7l1a*[m881/m881]/*tcf7l1b*[+/zf157tg] double mutants (Young and Wilson, unpublished).

## RNA extraction, reverse transcription and qPCR

Total RNA and genomic DNA were isolated from individual embryos at 10hpf following Life Technologies Trizol protocol. cDNA was synthesised by reverse transcription using SuperscriptII (Life Technologies) with 200 ng of total RNA to a final volume of 40 µl and oligo dT for priming. The cDNA reaction was diluted 10 times and 5 µl were used in 25 µl final volume reactions using GoTaq qPCR Master mix (Biorad). Each experimental condition was processed in technical and biological triplicates. All primers used had PCR efficiencies within 90–100% range: *gapdh* (F-ACC CGT GCT GCT TTC TTG, R-CTG CCT TAA CCT CAC CCT TG); *hprt1* (F-AAC AGT GAT CGC TCC ATT CC, R-GGA CAG ATC ATC TCC ACC AAT C); *lef1*(F-GCT TCA GGT ACA GGC CAG AG, R-AAA GAC GTC CGC TTT CCT CC); *otx1a* (F-GGT GTT TCT TGG CTT TGT GG, R-GGG CTT GCT TGA GGT ATG A); *otx2* (F-TAC ACG TCA ACG GGC TA A, R-CTC GTC TCT GGT TTC GAG GA); *rx3* (F-TCC GAG TAC AGG TGT GGT TCC, R-CTC CTG TCG CCG CCA TTT A); *six3b* (F-TGC CAA AAA CAG GCT TCA GCA, R-CTG ACA TGG AGC GCA GAC T); *tcf7l1a* (F- AGC ACA CGA ACG TAT CTC CA, R-GAG TCT TTA AGA GCC GCC GA); *tcf7* (F-TGC TGC CGT ATG AAC ACT TC, R-TCT CCT GCG TCT GAT GTC TG); *tcf7l1b* (F-GGC TAA AGT AGT GGC CGA GTG, R-CTG GCC AGC TCG TAG TAT TTG); *tcfl2* (F-GCC TCC GCC TAG ATC TGA AA, R-CTT GCC TTT TTG CAG CCT CC).

Wildtype and *tcf7l1a*<sup>*m881*</sup> mutant cDNA fragments spanning the *tcf7l1a* exon 7/8 border for DNA sequencing were amplified with primers P2 and P3 (*Kim et al., 2000*).

## RNA sequencing

Total RNA was extracted from zebrafish embryos at 80% epiboly by Trizol extraction and gDNA was genotyped for *tcf7l1a* to identify wildtype, heterozygous and homozygous embryos. RNA from six wild-type and six *tcf7l1a*<sup>-/-</sup>mutant embryos was DNase treated for 20 min at 37°C followed by addition of 1 ml 0.5M EDTA and inactivation at 75°C for 10 min to remove residual DNA. RNA was then cleaned using 2 volumes of Agencourt RNAClean XP (Beckman Coulter) beads under the standard protocol. Stranded RNA-seq libraries were constructed using the Illumina TruSeq Stranded RNA protocol with oligo dT pulldown. Libraries were pooled and sequenced on two lanes of Illumina HiSeq 2000 in 75 bp paired-end mode. Sequence data were deposited in ENA under accession PRJEB9957. FASTQ files were aligned to the GRCz11 reference genome using TopHat (v2.0.13, options: –library-type fr-firststrand, *Kim et al., 2013*). The data were assessed for technical quality (GC-content, insert size, proper pairs etc.) using QoRTs (*Hartley and Mullikin, 2015*). Counts for genes were produced using htseq-count (v0.6.0 options: –stranded=reverse, *Anders et al., 2015*) with the Ensembl v93 annotation as a reference. Sequence data were deposited in ENA under accession PRJEB9957. Differential gene expression was analysed using DESeq2 (*Love et al., 2014*).

## *In situ* hybridisation, probe synthesis and tunel labelling

Whole mount in situ hybridisation was performed using digoxigenin (DIG) and fluorescein (FLU)-labelled RNA probes according to standard protocols (*Thisse and Thisse, 2008*). Probes were synthesized using T7 or T3 RNA polymerases (Promega) according to manufacturers' instructions and supplied with DIG or FLU labelled UTP (Roche). Probes were detected with anti-DIG-AP (1:5000, Roche), anti-FLU-AP (1:10000, Roche), or anti-DIG-POD (1:1000, Roche) antibodies and developed with NBT/BCIP mix (Roche), for regular microscopy or Fast Red (Sigma) or CY-3 tyramide (*Lauter et al., 2011*) substrate for confocal analysis.

For Tunel assays, embryos were processed as for *in situ* hybridisation up to the washing of PK stage. After this, embryos were incubated in an acetone:ethanol (2:1)–20°C prechilled solution at −20°C for 10 min. After PBS tween 0.5% washes, embryos were incubated for 1 hr in the equilibration buffer (Millipore ApopTag kit) at room temperature. The buffer was removed and 20 µl of fresh equilibration buffer was added plus 15 µl of TdT mix (12 µl reaction buffer, 6 µl TdT enzyme, 0.5 µl of 10% triton X100), and embryos incubated at 37°C over night. Samples were washed for 1 hr at 37°C and 1 hr at room temperature, and the protocol was continued as for *in situ* hybridisation.

## Quantification of eye profile, eye volume, cell volume, PH3 +cells, and posterior lateral line primordium (pLLP) position

The eye profile and eye volume were calculated from confocal imaging of *vsx2* in situ hybridisation stained embryos at 24hpf. The eye volume/eye profile ratio average from 10 embryos was 53.24. This ratio was used to estimate eye volume from eye profile area as the profile area to eye volume ratio is approximately constant after 24hpf (*Matejčić et al., 2018*). The sizes of eye profiles were quantified from lateral view images of PFA-fixed embryos by delineating the eye using Adobe Photoshop CS5 magic wand tool and measuring the area of pixels included in the delineated region. The surface area was then transformed from px$^2$ to µm$^2$.

For estimation of cell volume, 3d stacks were first contrast enhanced to increase the intensity of the labelled cells in the entire volume. Subsequently, a 3d median filter was applied to filter out high intensity noise. Next, a fixed threshold was applied to segment individual cells in the volume and their surface area and volume were calculated. For each imaged and analysed image stack, we also manually inspected the processed data to ensure that post processing did not result in partial segmentation of cell volumes. Cells that were undergoing division or were within ~20 µm from the dorsal or ventral surface of the imaged volume were excluded from the analysis. Image processing and analysis was carried out using ImageJ.

For PH3 quantification, embryos were oriented to yield a lateral view of the retina, and the widest plane of the retina was imaged. The z-series was defined as 2 µm above and 2 µm below the widest plane of the retina. Stacks were taken at a step size of 1 µm for a total of 5 stacks per imaged

volume. The number of nuclei undergoing mitosis, marked by α-PH3, were counted. The total number of nuclei for each eye was estimated by counting the DAPI labeled nuclei in a standardized area of 40 × 40 pixels. This result was multiplied by the total area of the retina (in pixels), obtained using the freehand tool in ImageJ and the software's measuring capabilities, and then divided by 1600. To normalize the data, the number of α-PH3 positive cells was presented as a percentage of the total number of nuclei per retina.

pLLP migration was measured by analysing the position of the posterior end of the primordium relative to the somite boundary labelled by in situ hybridisation with eya1 and xirp2a respectively.

## Confocal microscopy and image analysis

Confocal imaging was performed on a Leica TCS SP8 confocal microscope. For time lapse analyses, the stage was set in an air chamber heated to 28.5°C. Live embryos were immobilized in 1% low melting point agarose (Sigma) and 0.016% Tricaine (Sigma) to anesthetize. Image volume analysis measurement was performed on Imaris 7.7.0 and Fuji.

## Cell transplantation

WIldtype or MZtcf7l1a⁻/⁻ embryos used as donors were injected with 50 pg of GFP mRNA at 1 cell stage. At 3-4hpf, blastula stage, dechorionated donor and host embryos were mounted in 3% methylcellulose in fish water supplemented with 1% v/v penicillin/streptomycin (5000 units penicillin and 5 mg streptomycin per ml) and viewed with a fixed-stage compound microscope (Nikon Optiphot). Approximately 30–40 cells were taken from the animal pole of donors and transplanted to approximately the same position in hosts by suction using an oil-filled manual injector (Sutter Instrument Company). Embryos were moved to 1% penicillin/streptomycin supplemented fish media and fixed at 10hpf.

## Eye vesicle cell removal

Embryos were mounted in 1% low melting point agarose in Ringer's solution supplemented with 1% v/v penicillin/streptomycin. A slice of set agarose was removed to expose one of the eyes and a drop of mineral oil (sigma) was placed over the target eye to dissolve the epidermis (*Picker et al., 2009*). After 2 min, the oil drop was removed and optic vesicle cells were sucked out with a capillary needle filled with mineral oil. Embryos were left to recover for half an hour before being released from the agarose.

## Optokinetic response

Optokinetic responses were examined using a custom-built rig to track horizontal eye movements (optokinetic nystagmus) in response to whole-field motion stimuli. Larvae at 4dpf were mounted in 1% low melting point agarose in fish water and analysed at 5dpf. The agarose surrounding the eyes was removed to allow normal eye movements. Sinusoidal gratings with spatial frequencies of 0.05, 0.1, 0.13 and 0.16 cycles/degree were presented on a cylindrical diffusive screen 25 mm from the centre of the fish's head with a MicroVision SHOWWX + projector. Gratings had a constant velocity of 10 degrees/s and changed direction and/or spatial frequency every 20 s. Eye movements were tracked under infrared illumination (720 nm) at 60 Hz using a Flea3 USB machine vision camera controlled with custom-written LABVIEW software. MATLAB scripts were used to extract slow phase eye velocity from recorded eye position data (degrees per second).

## Acknowledgements

We are grateful to Dr. Ajay Chitnis and Dr. Richard Dorsky for sharing reagents and fish lines, Dr. Bill Harris and colleagues for support and discussions, Lisa Tucker for technical assistance, the UCL fish facility for fish care, and Elke Ober participating in the genetic screen. We would like to thank Neha Wali and the Wellcome Sanger Institute sequencing pipelines for performing sequencing. This work was generously supported by funding from a Marie Curie Incoming International Fellowship to RY, Wellcome Trust grants to SW and RY (088175, 104682/Z/14/Z), and EB (WT098051, 206194), an MRC Programme grant to SW and GG (MR/L003775/1), Royal Society International Joint Project funding to SW and MA and FONDAP (15090007) to MA.

## Additional information

### Funding

| Funder | Grant reference number | Author |
|---|---|---|
| Wellcome Trust | 088175/Z/09/Z | Rodrigo M Young Stephen W Wilson |
| H2020 Marie Skłodowska-Curie Actions | Marie Curie Incoming International Fellowship | Rodrigo M Young |
| Royal Society | | Miguel L Allende Stephen W Wilson |
| Fondo de Financiamiento de Centros de Investigación en Áreas Prioritarias | 15090007 | Miguel L Allende |
| Wellcome Trust | WT098051, 206194 | Elisabeth M Busch-Nentwich |
| Medical Research Council | MR/L003775/1 | Gaia Gestri Stephen W Wilson |
| Wellcome Trust | 104682/Z/14/Z | Stephen W Wilson |
| Wellcome Trust | 089227/Z09/Z | Stephen W Wilson |

The funders had no role in study design, data collection and interpretation, or the decision to submit the work for publication.

### Author contributions

Rodrigo M Young, Conceptualization, Data curation, Formal analysis, Supervision, Investigation, Methodology, Writing—original draft, Writing—review and editing, Conceived the project, led and performed the genetic screen together with FC, TH, HS, QS, LL and CW; analysed the data and wrote the manuscript together with SW with input from all co-authors but primarily from FC, HS, TH, QS and GG; Thomas A Hawkins, Data curation, Formal analysis, Investigation, Methodology, Writing—review and editing, Performed the genetic screen together with RY, FC, HS, QS, LL and CW; Florencia Cavodeassi, Data curation, Formal analysis, Investigation, Methodology, Writing—review and editing, Performed the genetic screen together with RY, TH, HS, QS, LL and CW; Heather L Stickney, Data curation, Formal analysis, Investigation, Methodology, Writing—review and editing, Performed the genetic screen together with RY, TH, FC, QS, LL and CW; Quenten Schwarz, Investigation, Writing—review and editing, Performed the genetic screen together with RY, TH, FC, HS, LL and CW; Lisa M Lawrence, Investigation, Performed the genetic screen together with RY, TH, FC, HS, QS, and CW; Claudia Wierzbicki, Investigation, Performed the genetic screen together with RY, TH, FC, HS, QS and LL; Bowie YL Cheng, Jingyuan Luo, Elizabeth Mayela Ambrosio, Allison Klosner, Jasmine Rowell, Chintan A Trivedi, Isaac H Bianco, Investigation; Ian M Sealy, Elisabeth M Busch-Nentwich, Data curation, Investigation; Miguel L Allende, Funding acquisition; Gaia Gestri, Investigation, Writing—review and editing; Stephen W Wilson, Conceptualization, Supervision, Methodology, Writing—original draft, Project administration, Writing—review and editing

### Author ORCIDs

Rodrigo M Young http://orcid.org/0000-0001-5765-197X
Thomas A Hawkins https://orcid.org/0000-0003-2921-0004
Florencia Cavodeassi https://orcid.org/0000-0003-4609-6258
Quenten Schwarz http://orcid.org/0000-0002-5958-4181
Claudia Wierzbicki https://orcid.org/0000-0001-7266-2597
Bowie YL Cheng http://orcid.org/0000-0003-0794-6133
Elizabeth Mayela Ambrosio http://orcid.org/0000-0001-7227-7744
Ian M Sealy https://orcid.org/0000-0002-2890-6635
Jasmine Rowell http://orcid.org/0000-0001-7040-8528
Chintan A Trivedi http://orcid.org/0000-0003-3890-0744
Isaac H Bianco https://orcid.org/0000-0002-3149-4862

Miguel L Allende ⓘ http://orcid.org/0000-0002-2783-2152
Elisabeth M Busch-Nentwich ⓘ http://orcid.org/0000-0001-6450-744X
Gaia Gestri ⓘ http://orcid.org/0000-0001-8854-1546
Stephen W Wilson ⓘ https://orcid.org/0000-0002-8557-5940

### Ethics

Animal experimentation: This study was performed in strict accordance to the recommendations of the Home Office, UK.

### Decision letter and Author response

Decision letter https://doi.org/10.7554/eLife.40093.033
Author response https://doi.org/10.7554/eLife.40093.034

## Additional files

### Supplementary files

• Supplementary file 1. Supplementary tables. (**A**) RT-qPCR data on mRNA levels from $Ztcf7l1a^{-/-}$ versus wildtype embryos. Figures represent the fold change of $Ztcf7l1a^{-/-}$ over wildtype RT-qPCR experiments performed as technical and biological triplicates using GAPDH as a reference control. Standard deviation (SD). p-Value calculated by an unpaired t-test. (**B**) Measurement of the area of the prospective forebrain in wildtype and $Ztcf7l1a$ mutants. Prospective forebrain size data generated by measuring the area enclosed by $emx3$ expression to the rostral limit of $pax2a$ (mesencephalic marker) expression after whole mount *in situ* hybridisation in wildtype (+/+), $tcf7l1a^{+/-}$ (+/-) and $Ztcf7l1a^{-/-}$ (-/-) embryos at 10hpf. Data in $\mu m^2$. Average (avg), standard deviation (SD). Not-significant, ns. p-Value calculated by an unpaired t-test. (**C**) Measurement of the volume of $rx3$ expression in the eye field by fluorescent *in situ* hybridisation in wildtype and $tcf7l1a$ mutants. Confocal volume analysis of $rx3$ fluorescent *in situ* hybridisation on wildtype (+/+), $tcf7l1a^{+/-}$ (+/-) and $Ztcf7l1a^{-/-}$ (-/-) at 10hpf. Data in $\mu m^3$. Average (avg), standard deviation (SD). Not-significant (ns). p-Value calculated by an unpaired t-test. (**D**) List of differentially expressed genes between wildtypes and $tcf7l1a^{-/-}$, and between $tcf7l1a^{-/-}$ and $tcf7l1a^{-/-}tcf7l1b^{+/-}$ double mutants at 8hpf. Ensembl gene ID (Gene ID), p-value (pval), adjusted p-value (adjp), log2 fold change (log2fc), linear fold change (fold change), chromosome (Chr) and gene name (Name). (**E**) Optokinetic response analysis of wildtype and $Ztcf7l1a^{-/-}$ mutants. Tabulation of the data used on the plot (*Figure 4—figure supplement 1*). No significant difference observed when performing an unpaired t-test. Average (avg), standard deviation (SD). (**F**) Eye volume measurement data in wildtype and $tcf7l1a$ mutants. Data of eye volume measurement ($\mu m^3$) from wildtype and $tcf7l1a^{-/-}$ embryos at 24, 28, 32, 36, 48, 60, 72 and 96hpf. Average (avg), standard deviation (SD), percentage of $Ztcf7l1a^{-/-}$ eye volume size relative to wildtype eyes (%). (**G**) Quantification of the posterior lateral line primordium position in wildtype and $Ztcf7l1a^{-/-}$ mutants. Tabulation of the data used on the plot (*Figure 4—figure supplement 2*). No significant difference observed when performing an unpaired t-test. Average (avg), standard deviation (SD). (**H**) Cell volume quantification data in $tcf7l1a^{-/-}$ mutants and sibling eyes. Data of eye cell volume measurement ($\mu m^3$) in siblings and $tcf7l1a^{-/-}$ mutants at 24 and 36hpf. In sibling columns, wildtype cell data is in bold text and heterozygotes in normal text. Average (Avg), standard deviation (SD). (**I**) Eye volume measurement data from eye vesicle cell-removed embryos. Data of eye volume measurement ($\mu m^3$) from control and optic-vesicle ablated eyes at 30, 36, 54, 78 and 102hpf. Percentage of ablated eye volume size relative to the control eye (%). The last time point is missing for embryos 1 and 9 because they died after 78 hr. (**J**) Classification and quantification of $atoh7$ expression patterns in wildtype and $Ztcf7l1a^{-/-}$ eyes. Quantification of $atoh7$ expression categories in the eye at 28, 32, 36, 40, 44, 48 and 52hpf in wildtype (top table) and $Ztcf7l1a^{-/-}$ (bottom table) embryos. $atoh7$ expression was classified in the following categories: VN, ventro nasal; VN+, ventro nasal plus a few scattered cells; N+, nasal plus scattered cells covering the whole retina; NR, nasal retina; WR, whole retina; PR, peripheral retina. (**K**) Eye volume measurement data in heat-shocked control wildtype and $Tg(HS:dkk1)^{w32}$ embryos. Tabulation of eye volume measurements in $\mu m^3$ in heat shock control wild-type and $Tg(HS:dkk1)^{w32}$ embryos at 28, 32, 36, 48, 60 and 72 hpf, used for plot in *Figure 5R*. Average (avg), standard deviation (SD), percentage of $Tg(HS:dkk1)^{w32}$ eye volume

size relative to heat shock control wildtype eyes (%). (**L**) Phosphohistone3 (PH3+) and double *atoh7*/ PH3+ cell counts in wildtype and Z*tcf7l1a* mutant eyes. Tabulation of the PH3+ and double *atoh7*/ PH3+ cell count in wildtypes (A) and Z*tcf7l1a*$^{-/-}$ mutants (B). The percentage of total PH3+ cells was calculated by dividing the PH3 count by the total number of cells. The percentage of nasal (N), temporal (T) or whole retina (total) *atoh7*/PH3+ cells was calculated by dividing the *atoh7*/PH3 +cell count in the N, T or total retina by the PH3+ count in the respective area. Average, avg; Standard deviation, SD. (**M**) List of genes in the *U910* genetic interval. Position of the genes in the *U910* interval in megabases (Mb) in the GRCz10 assembly. Mapped gene is highlighted in yellow. (**N**) Frequency of eyeless embryos and their respective genotypes in four incrosses of *hesx1*$^{-/-}$/ *tcf7l1a* $^{\pm}$ fish. (**O**) Eye volume measurement data in wildtype, Z*tcf7l1a*, *cct5* and *cct5/tcf7l1a* double mutant siblings. Data of eye volume measurement figures in μm$^3$ in embryos at 36 (A) and 52hpf (B-D), injected with 2 pmol of *tp53* morpholino (C) or 0.8 pmol of *cct3* morpholino (D). Average (avg), standard deviation (SD). (**P**) List of the genes in the *U762* genetic interval. Position of genes in the *U762* interval in megabases (Mb) in the GRCz10 assembly. Mapped gene is highlighted in yellow. (**Q**) PH3 and double *atoh7*/PH3 positive cell count in *cct5*/Z*tcf7l1a*$^{-/-}$ mutants. Tabulation of the PH3+ (A) and double *atoh7*/PH3+ (B) cell count in wildtype, Z*tcf7l1a*$^{-/-}$, *cct5*$^{-/-}$ and *cct5*/Z*tcf7l1a*$^{-/-}$ eyes. Avg, average; SD, standard deviation. (**R**) Eye volume measurement data in wildtype, Z*tcf7l1a*, *gdf6a* and *gdf6a/Ztcf7l1a* double mutant siblings. Data of eye volume measurement figures in μm$^3$ in embryos at 36hpf. Average (avg), standard deviation (SD). (**S**) Eye field volume measurement data in wildtype, Z*tcf7l1a*, *gdf6a* and *gdf6a/Ztcf7l1a* double mutant siblings. Data of eye field volume quantification in μm$^3$ from *rx3 in situ* hybridisation in wildtype, Z*tcf7l1a*$^{-/-}$, *gdf6a*$^{-/-}$ and *gdf6a/Ztcf7l1a* double mutant siblings at 10hpf. Average (avg), standard deviation (SD).
DOI: https://doi.org/10.7554/eLife.40093.028

• Transparent reporting form
DOI: https://doi.org/10.7554/eLife.40093.029

## Data availability

All the data used for this study was provided in the uploaded manuscript.

The following dataset was generated:

| Author(s) | Year | Dataset title | Dataset URL | Database and Identifier |
|---|---|---|---|---|
| Wellcome Sanger Institute | 2016 | Transcriptome_profiling_of_ zebrafish_tcf3a_and_tcf3b_ knockout_embryos | https://www.ebi.ac.uk/ ena/data/view/ PRJEB9957 | European Nucleotide Archive, PRJEB9957 |

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
