## [Decision Letter]

Thank you for submitting your article "Compensatory mechanisms render Tcf7l1a dispensable for eye formation despite its requirement in eye field specification" for consideration by *eLife*. Your article has been reviewed by three peer reviewers, and the evaluation has been overseen by Deborah Yelon as the Reviewing Editor and Marianne Bronner as the Senior Editor. The following individuals involved in the review of your submission have agreed to reveal their identity: Shawn Burgess (Reviewer #2); Juan R Martínez-Morales (Reviewer #3).

The reviewers have discussed their reviews with one another and the Reviewing Editor has drafted this decision to help you to prepare a revised submission.

Summary:

In this manuscript, Young, Wilson, and colleagues identify compensatory mechanisms at the cellular and genetic levels that contribute to the variable expressivity of *tcf7l1a* mutations on eye size in zebrafish. They demonstrate that compensatory eye size mechanisms function more broadly than that related to the specific *tcf* mutation, by manipulating eye field size through cell ablation and genetic-based cell expansion. These data suggest the existence of a feedback mechanism that can sense eye organ size and compensate growth/differentiation rates in order to generate a properly sized eye. The authors were unable to detect direct gene compensation by upregulation of specific *tcf* family members. Through an additional genetic screen, they isolated three genetic modifiers in a *tcf7l1a*^-/-^ background, one of which appears to be a bona fide synthetic interaction (but had also been previously described) and the other two of which appear to represent additive genetic interactions. Altogether, this is a very interesting study that provides a meaningful contribution to the emerging dialog in the community about network robustness in relation to patterning and development, and this topic should have broad appeal to the *eLife* readership. However, it would be helpful to provide additional experimental support for some of the main conclusions of the manuscript and to clarify some of the authors' interpretations, as described below.

Essential revisions:

1) The authors show compelling evidence of compensatory growth during eye development. This is a very interesting observation, and the mechanisms responsible for it are intriguing. The authors explain proliferative growth as a consequence of delayed neurogenesis. To support this, *atoh7* expression is shown in Figure 5. Although informative, measurement of neurogenesis is an indirect way of assessing proliferation, and thus it is insufficient to fully support claims on proliferative growth. Direct experiments investigating proliferation rate (PH3, BrdU pulses), and maybe also cell death (TUNEL or caspase stainings) and cell size (this is unlikely, but it is still a possibility to explain growth), should yield key information on the nature of the compensatory mechanisms. Of particular importance will be to characterize whether or not specific retinal regions are responsible for the compensatory growth, both in *tcf7l1a* mutants and upon retinal partial ablation.

2) Given the cell death associated with *cct5* mutations, it would be important to know whether eye size might be rescued by blocking apoptosis (such as with a p53 MO or expression of a bclxL transgene). Knowledge of this result will have an impact on interpreting the cellular mechanisms for compensatory eye size regulation (i.e. cell proliferation or cell survival)? A more detailed characterization on cell cycle kinetics would also be informative for this mutation. In addition, do the authors know whether other *cct* knockdowns or knockouts have the same additive phenotype as *cct5*?

3) The authors have carried out a valuable genetic screen for *tcf7l1a* modifiers. The screen is technically correct, but the way that synthetic phenotypes are assessed, by morphological observation at relatively late stages of development (48-72 hpf), opens the possibility of epistatic interactions taking place during very different developmental windows. The fact that the *tcf7l1a* mutation only displays early eye defects does not preclude the possibility of the gene playing a role later during development. In fact, the Wnt/β-Catenin pathway has been shown to play a role not only during eye field specification, but also in RPE specification, and CMZ differentiation and regenerative capacity. Whereas it is likely that *hesx1* may play a synergic role with *tcf7l1a* in eye field specification, this is less clear for *gdf6a* and even less for *cct5*. Additional experiments exploring the expression of eye field markers (either by qPCR or ISH: such as those in Figure 2) in double mutants for *tcf7l1a* and its modifiers (*hesx1, cct5* and *gdf6a*) would help to characterize the nature of the epistatic interactions, unveiling additional roles for *tcf7l1a* later during eye development.

4) Can the authors establish by transcriptional profiling what compensatory mechanisms are occurring in the outcrossed allele of *tcf7l1a* that does not occur in the original background?

Related to this, spot checking specific genes is a fairly limited way to find genetic compensation, but does profiling allow the authors to see network changes comparing mutant to wild-type siblings?

5) It is a bit of a concern that there were so many eyeless phenotypes seen in the initial cross after mutagenesis. Does it worry the authors that the genetic interactions from the sensitized background are creating a sort of false positive group? That is, is the genetic interaction pointing to mechanisms of compensation or merely illuminating the loss of robustness in a non-specific way? It would be helpful if the authors could comment on this in their revised manuscript.

6) In addition, the authors should re-examine some of the statements in the text. Although retinal size may affect neurogenesis, at least in some genetic/experimental conditions, the opposite is not necessarily true. There are a number of examples in the literature showing that impaired neurogenesis does not affect eye size. Therefore, the authors may reconsider certain sentences, such as "the timing of the spread of neurogenesis across the retina is coupled to size of the eye", that could be misleading. Similarly, is delayed neurogenesis the causative mechanism for size recovery? Couldn't this be just a correlative epiphenomenon? Sentences such as: "we find that *tcf7l1a* mutant optic vesicles delay neurogenesis to enable size recovery" seem premature, as the current experimental evidence does not support direct causality.

7) The authors should also carefully consider their use of the words "compensatory" and "compensation". Although "compensatory" appears in the title, the authors use it with two different meanings: "genetic compensation", in the Stainier sense of the term, and "compensatory growth", to imply uncharacterized developmental checkpoints controlling organ size. The title seems to point more to the second concept. Maybe the authors should have used "compensatory growth" instead of "compensatory mechanisms" to distinguish both phenomena. As eye field specification is affected in *tcf7l1a* mutants even in the "permissive genetic background", I think that this indicates an incomplete "genetic compensation", which would be very complex to characterize. The other aspect of the work, size compensation in vertebrates, is a very interesting and poorly explored phenomenon that deserves further attention.

8) The authors should report the "n" and experimental replicates for each study presented (preferably in the figure Legends).

[Editors' note: further revisions were requested prior to acceptance, as described below.]

Thank you for resubmitting your manuscript entitled "Compensatory growth renders Tcf7l1a dispensable for eye formation despite its requirement in eye field specification" for further consideration at *eLife*. Your revised article has been favorably evaluated by Marianne Bronner (Senior Editor), a Reviewing Editor, and 3 reviewers. As you will see in the reviews reprinted below, all of the reviewers agree that the manuscript has been improved, and there are only two remaining issues, raised by Reviewer #1, that should be considered before acceptance. Could you please evaluate whether the Figure 8 legend needs to be adjusted? In addition, could you consider including the *tcf7l1a*^-/-^/tcf7l1b^+/-^ data, as encouraged by Reviewer #1? Once you have addressed these two points, we will be ready to move forward toward acceptance of your manuscript.

Reviewer #1:

The authors have sufficiently addressed my concerns. The additional data and commentary improves this strong manuscript.

I did find a type-o: Figure 8 legend, "0.8pmol cct3 MO" is listed, but I believe this should state "2.0 pmol p53 MO".

Regarding the inclusion of *tcf7l1a*^-/-^/tcf7l1b^+/-^ data (100% penetrance of eyeless phenotype, even with a 40% increase of tcfl1b is interesting. I favor describing these data in the final version and disagree with the authors that this does not fit within the scope of the manuscript. I think it does fit and adds depth.

Reviewer #2:

The authors have in my opinion adequately addressed any concerns I and the other reviewers had with the manuscript. The manuscript is significantly improved and ready for the next steps.

Reviewer #3:

This is certainly an improved version of the work. The new data provided adequately address all my previous concerns. I do not have further issues, thus I feel the manuscript is now ready for publication

---

## [Author Response]

Essential revisions:1) The authors show compelling evidence of compensatory growth during eye development. This is a very interesting observation, and the mechanisms responsible for it are intriguing. The authors explain proliferative growth as a consequence of delayed neurogenesis. To support this, atoh7 expression is shown in Figure 5. Although informative, measurement of neurogenesis is an indirect way of assessing proliferation, and thus it is insufficient to fully support claims on proliferative growth. Direct experiments investigating proliferation rate (PH3, BrdU pulses), and maybe also cell death (TUNEL or caspase stainings) and cell size (this is unlikely, but it is still a possibility to explain growth), should yield key information on the nature of the compensatory mechanisms. Of particular importance will be to characterize whether or not specific retinal regions are responsible for the compensatory growth, both in tcf7l1a mutants and upon retinal partial ablation.

We agree with these points. To strengthen the idea that continued proliferation contributes to compensatory growth, we have added new data to the new Figure 6 and to Supplementary file 1K showing that at 36hpf, there are more phosphohistone3 positive (PH3+) cells in *tcf7l1a* mutants compared to wildtypes. We chose 36hpf for this assay because at this stage, there is a clear difference in the expression of *Tg(atoh7:RFP)* between mutants and wildtypes. At this stage, there is no significant bias in the number of PH3+ cells between the nasal and temporal retina (Figure 4—figure supplement 4). We also observed that a lower proportion of Ph3+ progenitor cells are committed to differentiation in *tcf7l1a* mutant compared to wildtype eyes as assessed by co-expression of *Tg(atoh7:RFP)* which labels cells in their final round of cell division before exiting the cell cycle (Zolessi et al., 2006; Figure 6A, B, F and Supplementary file 1K). In these experiments, there were more double-labelled cells in nasal than temporal retina (Figure 6F) as expected from the nasal to temporal progression of differentiation across the eye.

We quantified retinal cell volume at 24 and 36hpf (Figure 4—figure supplement 3) and Tunel staining at 36 and 48hpf (Figure 8—figure supplement 2) and observed no significant differences between wildtypes and *tcf7l1a* mutant siblings in any of these assays. This suggests that neither cell volume nor apoptosis play a significant role in the compensatory growth of the eye.

2) Given the cell death associated with cct5 mutations, it would be important to know whether eye size might be rescued by blocking apoptosis (such as with a p53 MO or expression of a bclxL transgene). Knowledge of this result will have an impact on interpreting the cellular mechanisms for compensatory eye size regulation (i.e. cell proliferation or cell survival)? A more detailed characterization on cell cycle kinetics would also be informative for this mutation. In addition, do the authors know whether other cct knockdowns or knockouts have the same additive phenotype as cct5?

We agree that the role of apoptosis in the genetic interaction between *cct5* and *tcf7l1a* was not resolved. To address this, and provide more information on the dynamics of eye growth in *cct5/tcf7l1a* mutants, we quantified eye size at 36hpf and 52hpf in embryos from a *cct5/tcf7l1a* double heterozygous incross, and also at 52hpf subsequent to inhibiting apoptosis by knocking down *tp53* with antisense morpholinos and assessing Tunel labelling. We found that even after knocking down *tp53*, eyes in homozygous double mutant *cct5/tcf7l1a* embryos show little or no further growth after 36hpf (Figure 8E, J, K, Supplementary file 1O). This suggests that increased apoptosis is not the primary cause of the failure in compensatory eye growth in *cct5/tcf7l1a* double mutant eyes. Although *cct5/tcf7l1a* double mutant eyes are small, they have significantly increased numbers of PH3+ cells compared to *tcf7l1a* mutants (Figure 8L-P, Supplementary file 1Q) suggesting that there may be cell cycle defects in the mutants.

We also agree with the reviewers that our results did not address whether the function of *cct5* in modifying the phenotype of *tcf7l1a* is independent or related to its function in the TCP-1 chaperonin complex. To explore this, we have knocked down the *cct3* chaperonin subunit and show that eyes in double knockdown *cct3/tcf7l1a* embryos also fail to recover their size at 52hpf, similar to *cct5* mutants (Figure 8K, Supplementary file 1O). This suggests that it is the impairment of TCP-1 chaperonin function that is modifying the *tcf7l1a* mutant phenotype.

3) The authors have carried out a valuable genetic screen for tcf7l1a modifiers. The screen is technically correct, but the way that synthetic phenotypes are assessed, by morphological observation at relatively late stages of development (48-72 hpf), opens the possibility of epistatic interactions taking place during very different developmental windows. The fact that the tcf7l1a mutation only displays early eye defects does not preclude the possibility of the gene playing a role later during development. In fact, the Wnt/β-Catenin pathway has been shown to play a role not only during eye field specification, but also in RPE specification, and CMZ differentiation and regenerative capacity. Whereas it is likely that hesx1 may play a synergic role with tcf7l1a in eye field specification, this is less clear for gdf6a and even less for cct5. Additional experiments exploring the expression of eye field markers (either by qPCR or ISH: such as those in Figure 2) in double mutants for tcf7l1a and its modifiers (hesx1, cct5 and gdf6a) would help to characterize the nature of the epistatic interactions, unveiling additional roles for tcf7l1a later during eye development.

We agree with the reviewers that *tcf7l1a* may continue to play roles in eye development subsequent to eye field specification; indeed we did not mean to imply that the interactions between *tcf7l1a* and *cct5* or *gdf6a* were necessarily occurring at very early stages.

Measurements of eye size comparing wildtype with *tcf7l1a*, *cct5* and *cct5*/*tcf7l1a* mutants show that *cct5* single mutant 36hpf eyes have comparable size to wildtype eyes, and single *tcf7l1a* and double *cct5/tcf7l1a* mutants also have comparable eye size at this stage (Figure 8). This data suggests no significant role for *cct5* in early eye specification and implies the genetic interaction between *cct5* and *tcf7l1a* occurs at later developmental stages. Both *cct5* and *tcf7l1a* single mutant eyes continue to grow after 36h whereas double *cct5/tcf7l1a* mutants do not. This suggests the genetic interaction may be due to roles for both genes in retinal progenitor proliferation.

By 36hpf, eye size in *gdf6a/tcf7l1a* double mutants is already reduced to about half the volume of Z*tcf7l1a* mutant eyes (Figure 10A-D, M; Supplementary file 1R). This suggests that early steps in eye formation are compromised in these embryos. Much to our surprise given that *gdf6a* is not described as being expressed in the eye field (Rissi et at., 1995), we found that the eye field was smaller than wildtype in *gdf6a* single mutants and smaller in *gdf6a/tcf7l1a* double mutants than in *tcf7l1a* single mutants (Figure 10I-L, Supplementary file 1S). These results reveal a likely genetic interaction between *gdf6a* and *tcf7l1a* occurring surprisingly early in eye formation. We are aware that there is more to be done to further our understanding of the function of these genes in eye formation, but also think that this research is beyond the scope of the current study.

4) Can the authors establish by transcriptional profiling what compensatory mechanisms are occurring in the outcrossed allele of tcf7l1a that does not occur in the original background?Related to this, spot checking specific genes is a fairly limited way to find genetic compensation, but does profiling allow the authors to see network changes comparing mutant to wild-type siblings?

We have not maintained incrosses of the original line carrying the *tcfl1a* mutation and showing the eye loss phenotype and so cannot address this exact question. However, we add data from a related RNAseq experiment (Supplementary file 1D) comparing six individual wildtype and six Z*tcf7l1a* mutants replicate mRNA samples extracted at the initiation of eye specification (80% epiboly). We have added a paragraph to the third section of the paper describing the results. The data shows the expected down regulation of *tcf7l1a* and *rx3* in *tcf7l1a* mutants, but also other forebrain markers *fezf2* (telencephalon and diencephalon, Sun et al., 2006) and *hesx1*. The data also shows upregulation of genes including *her5* and *irx1b* (Müller et al., 1996, Wang et al., 2001), consistent with partial posteriorisation of the forebrain (subsection “Optic vesicles evaginate and form eyes in MZtcf7l1a^-/-^ mutants despite a much-reduced eye field.”; Supplementary file 1C). There is no obvious up or down regulation of a likely candidate gene that could compensate for the lack of *tcf71a.*

5) It is a bit of a concern that there were so many eyeless phenotypes seen in the initial cross after mutagenesis. Does it worry the authors that the genetic interactions from the sensitized background are creating a sort of false positive group? That is, is the genetic interaction pointing to mechanisms of compensation or merely illuminating the loss of robustness in a non-specific way? It would be helpful if the authors could comment on this in their revised manuscript.

We have added a comment in the results and a few lines to a paragraph to the Discussion mentioning the issues raised by the reviews.

6) In addition, the authors should re-examine some of the statements in the text. Although retinal size may affect neurogenesis, at least in some genetic/experimental conditions, the opposite is not necessarily true. There are a number of examples in the literature showing that impaired neurogenesis does not affect eye size. Therefore, the authors may reconsider certain sentences, such as "the timing of the spread of neurogenesis across the retina is coupled to size of the eye", that could be misleading. Similarly, is delayed neurogenesis the causative mechanism for size recovery? Couldn't this be just a correlative epiphenomenon? Sentences such as: "we find that tcf7l1a mutant optic vesicles delay neurogenesis to enable size recovery" seem premature, as the current experimental evidence does not support direct causality.

We agree with the reviewers and have moderated our text accordingly.

7) The authors should also carefully consider their use of the words "compensatory" and "compensation". Although "compensatory" appears in the title, the authors use it with two different meanings: "genetic compensation", in the Stainier sense of the term, and "compensatory growth", to imply uncharacterized developmental checkpoints controlling organ size. The title seems to point more to the second concept. Maybe the authors should have used "compensatory growth" instead of "compensatory mechanisms" to distinguish both phenomena. As eye field specification is affected in tcf7l1a mutants even in the "permissive genetic background", I think that this indicates an incomplete "genetic compensation", which would be very complex to characterize. The other aspect of the work, size compensation in vertebrates, is a very interesting and poorly explored phenomenon that deserves further attention.

We agree with the reviewers that “compensatory mechanisms” could be applied to both genetic and growth compensation. Although “compensatory growth” may not fully encompass everything we discuss, we do agree that it is less open to mis-interpretation and have changed the title of the manuscript accordingly.

8) The authors should report the "n" and experimental replicates for each study presented (preferably in the figure Legends).

The ‘n’s and experimental replicate numbers are now in the figure legends.

[Editors' note: further revisions were requested prior to acceptance, as described below.]

Reviewer #1:The authors have sufficiently addressed my concerns. The additional data and commentary improves this strong manuscript.I did find a type-o: Figure 8 legend, "0.8pmol cct3 MO" is listed, but I believe this should state "2.0 pmol p53 MO".Regarding the inclusion of tcf7l1a^-/-^/tcf7l1b^+/-^ data (100% penetrance of eyeless phenotype, even with a 40% increase of tcfl1b is interesting. I favor describing these data in the final version and disagree with the authors that this does not fit within the scope of the manuscript. I think it does fit and adds depth.

Based on the suggestions of reviewer#1 and have added a new dataset including the results of a RNAseq analysis of *tcf7l1a/tcf7l1b* double mutants (Supplementary file 1D) and also added a comment regarding this to the Results section of the manuscript. The suspected typo noted by this reviewer is not an error (Rev#1: “I did find a type-o: Figure 8 legend, "0.8pmol *cct3* MO" is listed, but I believe this should state "2.0 pmol p53 MO"). However, the *cct5*+*tp53* morpholino condition is already mentioned earlier in the same section of Figure legend 8; and the following section is correct in mentioning *cct5*+*cct3mo* as it refers to the data included in the bar plot in Figure 8K. We hope this is sufficient to answer the concerns of reviewer#1 and that you now consider that the manuscript is ready for publication.